# Effect of Bone Quality, Implant Length, and Loading Timing on Stress Transmission in the Posterior Mandible: A Finite Element Analysis

**DOI:** 10.3390/bioengineering12080888

**Published:** 2025-08-20

**Authors:** Ladise Ceylin Has, Recep Orbak

**Affiliations:** Department of Periodontology, Faculty of Dentistry, Atatürk University, 25240 Erzurum, Türkiye; rorbak@atauni.edu.tr

**Keywords:** finite element analysis, short dental implants, bone grafting, peri-implant stress, mandibular atrophy, loading protocols, implant biomechanics

## Abstract

This study aimed to evaluate the biomechanical effects of implant length, mandibular morphology, graft application, loading timing, and force direction on peri-implant stress distribution using finite element analysis (FEA). Five mandibular models representing normal, atrophic, and graft-augmented conditions were constructed. Each model was analyzed with 6 mm and 12 mm Straumann Standard implants under two loading types, vertical (200 N) and oblique (100 N at 30°), across three loading protocols (immediate, early, and delayed). Stress analysis was conducted using von Mises and principal stress criteria, focusing on cortical and trabecular bone, the implant–abutment complex, and the mandibular canal. Under vertical loading, increasing the implant length from 6 mm to 12 mm reduced the maximum tensile stresses in trabecular bone from 0.930 MPa to 0.475 MPa (an approximate 49% decrease). However, oblique loading caused a substantial increase in stresses in all regions, with trabecular compressive stress reaching up to −19.102 MPa and cortical tensile stress up to 179.798 MPa in the atrophic mandible. Graft application significantly reduced peri-implant stresses; for example, maximum compressive stress in the cortical bone decreased from −227.051 MPa in the atrophic model to −13.395 MPa in the grafted model under similar loading conditions. Although the graft donor site was not explicitly modeled, the graft material (Bio-Oss) was anatomically positioned in the posterior mandible to simulate buccolingual augmentation and its biomechanical effects. Stress concentrations around the mandibular canal remained below the 6 MPa threshold for neurovascular injury in all scenarios, indicating a biomechanically safe outcome. These findings indicate that oblique loading and reduced bone volume may compromise implant survival, whereas graft application plays a critical role in mitigating stress levels and enhancing biomechanical stability. The study also emphasizes the importance of considering force direction and bone quality in clinical planning, and highlights the novelty of combining graft simulation with FEA to assess its protective role beyond implant length alone.

## 1. Introduction

Over the past half-century, dental implants have revolutionized the management of edentulism, becoming a transformative approach in dental practice and prosthetic rehabilitation. With the widespread use of implant-supported prostheses, the restoration of functional efficiency and esthetic integrity in individuals with partial or total tooth loss has become significantly more predictable, patient-centered, and successful in the long term. This success is supported by extensive clinical data indicating high survival rates and favorable patient outcomes in implant-based therapy [1].

Following tooth loss, the alveolar resorption process leads to a notable decrease in bone height, particularly in the anterior and posterior segments of the mandible. This reduction limits the availability of sufficient vertical and horizontal bone volume for implant placement, complicates surgical planning, and increases the risk of complications.

As the alveolar ridge thins, the reduced distance between the crest and the inferior alveolar nerve considerably increases the likelihood of neurovascular complications such as nerve injury and paresthesia [2,3].

In cases with limited alveolar bone volume, the use of short implants is increasingly preferred to avoid invasive surgical procedures. Short implants offer the advantage of being placed with minimal intervention into the existing bone, without the need for grafting or advanced augmentation. This approach provides significant clinical benefits by reducing morbidity, especially in medically compromised or high-risk patients [4].

In clinical scenarios with insufficient bone volume, alveolar ridge augmentation is also commonly employed prior to implant placement. The aim of this approach is to provide adequate bone volume in the vertical and/or horizontal dimensions to enable stable placement of standard or long implants. Graft-induced bone volume gain makes it possible to use longer implants, which helps distribute occlusal loads over a wider area and promotes more balanced peri-implant stress distribution. However, these surgical techniques may increase procedure time, necessitate additional surgical sites, and carry the risk of donor site morbidity. Additionally, biological integration of the graft requires time [5,6].

The biomechanical interaction between the dental implant and surrounding bone is one of the most critical factors determining treatment success. Design parameters such as implant geometry, length, diameter, thread configuration, and surface roughness directly influence the magnitude and distribution of stress in the bone. The literature shows that implant angulation and physical properties can lead to microscopic fractures, localized bone resorption, and, ultimately, implant failure in the peri-implant region [7]. In addition to physical properties, the applied loading protocol significantly affects biomechanical forces and the osseointegration process. Loading time not only determines stress magnitude but also impacts the biological response to that stress. Given the rigid nature of cortical bone and the more elastic nature of trabecular bone, the localization of stress varies depending on the type of loading applied [8].

Direct and non-invasive evaluation of implant–bone interactions is not feasible in clinical settings due to technical and ethical limitations. Therefore, finite element analysis (FEA) has become a widely accepted and powerful modeling technique in dental implantology to investigate biomechanical behavior [9,10].

In this study, the biomechanical effects of implant length and loading protocols were numerically analyzed using the finite element method on models of normal, atrophic, and grafted atrophic mandibles with regard to osseointegration, fracture resistance, post-loading resorption, and accumulated stress in the bone.

The primary objective of the study was to comparatively assess the biomechanical performance of short and long implants under various loading scenarios in different mandibular morphologies, and thereby generate objective data to inform clinical decision-making and support individualized implant treatment protocols.

To guide this analysis, the following hypotheses were formulated:

Null hypothesis (H_0_): Variations in mandibular morphology, implant length, and loading protocol do not significantly affect peri-implant stress distribution.

Alternative hypothesis (H_1_): These anatomical and procedural factors have a significant influence on stress transmission specifically defined as follows:Atrophic mandibles result in higher peri-implant stress concentrations than normal or grafted models;Long implants reduce marginal stress and distribute occlusal forces more evenly than short implants;Delayed loading generates more favorable biomechanical outcomes by aligning force application with osseointegration.

## 2. Materials and Methods

This study was conducted as a collaborative project between the Faculty of Dentistry at Atatürk University and Tinus Technologies. Ethical approval for the study was obtained from the Ethics Committee of the Faculty of Dentistry at Atatürk University (Approval Date: 11 October 2024; Decision No: 61; Session No: September 2024).

The modeling and simulation workflow consisted of arranging the three-dimensional geometric network structure, converting it into a solid model suitable for finite element analysis (FEA), and performing mechanical stress analyses. All procedures were carried out on HP workstations equipped with INTEL Xeon E-2286 processors running at 2.40 GHz and 64 GB of ECC memory.

The primary bone structure was modeled using tomographic data sourced from the Visible Human Project (U.S. National Library of Medicine, Maryland, USA), which is a publicly available and anonymized anatomical dataset and thus exempt from requiring individual consent. Three-dimensional reconstructions of bone and dental structures were generated using the 3DSlicer software (version 5.8.1), from which surface models were exported in .stl format.

Reverse engineering processes and three-dimensional CAD modeling were performed in Blender software (version 4.4). The adaptation of these models for finite element simulation and mesh optimization was carried out using ALTAIR HyperMesh 2022. Finally, finite element analysis of the prepared models was conducted in ALTAIR OptiStruct 2022 (ALTAIR, Troy, MI, USA) (Figure 1).

### 2.1. Modeling of Cortical Bone, Trabecular Bone, and Teeth

The mandibular bone model used in this study was generated from tomographic data obtained through the Visible Human Project. These images were reconstructed with a slice thickness of 0.33 mm and saved in DICOM (.dcm) format. The data were imported into 3DSlicer software for three-dimensional visualization and segmentation.

Threshold-based segmentation was performed using Hounsfield Unit (HU) values ranging approximately from 426.50 to 3193.04. Non-bone tissues and artifacts were manually eliminated using tools such as “Erase” and “Scissors.” Accurate HU thresholding was crucial for isolating mineralized bone while excluding soft tissues. After segmentation, 3D volumetric models were generated and exported in .stl format.

These surface models were then imported into Blender software, where three distinct mandibular configurations were constructed: normal, grafted, and atrophic. The anatomical measurements for each model were derived from the vertical distances relative to the mandibular canal and are presented in Table 1.

The bone quality classification into D2, D3, and D4 types followed the well-established system introduced by Lekholm and Zarb [11]. In this scheme, the following definitions apply:D2 corresponds to dense cortical bone with coarse trabecular bone (normal mandible);D3 indicates thinner cortical bone with a finer trabecular structure (grafted mandible);D4 represents minimal cortical bone and very fine trabeculae (atrophic mandible).

This classification is widely used in implant planning due to its clinical relevance in describing local bone density and structure. It also provides a biomechanical basis for defining material properties in finite element models, ensuring that bone behavior is realistically simulated under loading.

All finalized 3D models were geometrically aligned using Blender’s coordinate system to ensure positional consistency across analyses.

### 2.2. Modeling of Implants, Abutments, Screws, Mandibular Canal, and Graft: Creation of Study Models

The 3D CAD models of two Straumann Standard dental implants (6 mm and 12 mm in length, both 4.1 mm in diameter) were created using Blender software, based on precise geometric data available from the manufacturer’s technical catalog. Corresponding abutments and fixation screws (10 mm in length and 2 mm in diameter) were also modeled in accordance with manufacturer specifications to ensure realistic assembly conditions.

The mandibular canal was modeled as a 1.5 mm diameter cylinder surrounded by a 0.25 mm alveolar nerve sheath, created using CAD tools in Blender. Additionally, the graft was designed according to the morphometric dimensions of the D3-type mandible (Table 1), and its geometry was integrated into the grafted bone model to reflect anatomical reality.

A feldspathic porcelain crown with the morphological features of a natural first mandibular molar was designed and placed atop each implant, ensuring consistent occlusal loading. The crown–abutment–implant interface was constructed as a continuous assembly with compatible mesh topology to allow accurate stress transmission across interfaces.

Mesh matching and optimization of all components (implant, abutment, screw, crown, bone, graft) were conducted using ALTAIR HyperMesh 2022 to ensure nodal continuity and prevent stress singularities in interface zones.

In total, five distinct finite element models were created by varying bone morphology, implant length, and bone quality. All implants were placed at the mandibular right first molar region (tooth #46). Model configurations were defined as shown in Figure 2, Figure 3, Figure 4, Figure 5 and Figure 6.

These five models were comparatively analyzed to assess the biomechanical effects of implant length and mandibular morphology on peri-implant stress distribution and load transfer patterns.

### 2.3. Material Properties

All structures in the models were assumed to exhibit isotropic, linearly elastic, and homogeneous mechanical behavior. The elastic modulus (Young’s modulus) and Poisson’s ratio of each material were defined based on values reported in previous biomechanical studies.

Trabecular bone was modeled with different elastic moduli according to bone quality classification:D2: 700 MPa;D3: 350 MPa;D4: 150 MPa;Poisson’s ratio for all trabecular bone types was taken as 0.30.

These values are in line with the data reported in the literature [12,13].

Cortical bone was modeled with an elastic modulus of 13,700 MPa and a Poisson’s ratio of 0.30 [14,15].

The graft material (Bio-Oss) was defined with an elastic modulus of 1370 MPa and a Poisson’s ratio of 0.30, based on values from the literature evaluating its biomechanical integration properties [16].

Titanium (used for both the implants and abutments) was assigned a Young’s modulus of 110,000 MPa and a Poisson’s ratio of 0.33, in accordance with widely accepted mechanical property data for Grade IV titanium [16].

The feldspathic porcelain used in the crown restoration was defined with an elastic modulus of 82,800 MPa and a Poisson’s ratio of 0.35, as described in previous studies on dental ceramic materials [17].

The mandibular canal, due to its soft tissue characteristics and relatively low elasticity, was modeled with an elastic modulus of 1.3 MPa and a Poisson’s ratio of 0.40 [18].

All materials were modeled as isotropic, linear-elastic, and homogeneous. This simplification was adopted to reduce computational complexity and ensure consistent comparisons across different loading and anatomical scenarios.

### 2.4. Generation of Mathematical Models

After completion of the geometric modeling in Blender, all models were transferred to ALTAIR HyperMesh 2022 for meshing and preparation for finite element analysis (FEA). To achieve high accuracy and stability in the simulations, each surface was meshed using triangular (tria) elements with a refined element size ranging between 0.1 and 0.25 mm, enabling the accurate representation of complex anatomical structures. Internal volumes were discretized using tetrahedral (solid) elements, which are well-suited for capturing the biomechanical response of heterogeneous biological materials.

This high-resolution meshing approach was selected to minimize numerical artifacts and improve stress resolution, particularly in cortical–trabecular interfaces and implant–bone contact zones (Figure 7). Previous studies have shown that finer mesh densities contribute to more accurate stress and strain predictions in maxillofacial FEA models [12,14].

The prepared mathematical models were exported to the ALTAIR OptiStruct 2022 solver for analysis. In total, thirty simulations were performed—ten linear static and twenty nonlinear static analyses—across five model variants. This simulation count is notably higher than that in typical FEA studies in implant biomechanics, which often report only one or two load cases per model [16]. The use of multiple simulations under varying conditions improves the robustness and generalizability of the findings, allowing for a more nuanced understanding of how implant length and mandibular morphology affect biomechanical stress distribution.

### 2.5. Loading Scenarios and Boundary Conditions

To simulate physiological masticatory loading conditions in the posterior mandible, two types of force were applied (Figure 8):

A vertical load of 200 N was applied to the central fossa of the crown on tooth #46.

A 30° oblique load of 100 N was applied to the lingual slope of the buccal cusp of the same crown.

This dual-loading protocol mimics realistic occlusal forces and is consistent with FEA studies in dental biomechanics [19,20,21].

The loads were distributed evenly across multiple surrounding nodes to avoid the formation of artificial stress concentrations (singularities) at the point of application, which would compromise the accuracy of the simulation.

To ensure structural stability and eliminate rigid body motion, all degrees of freedom were fully constrained at selected superior and inferior cortical and trabecular bone nodes in all three translational and rotational axes (Figure 9).

To prevent stress singularities, the applied forces were equally distributed among the surrounding nodes in the loading regions. The boundary conditions were defined on the mesial, distal, and basal surfaces of the mandibular model. At these regions, the displacement degrees of freedom (Ux, Uy, Uz) of all nodes in the three axes (X, Y, Z) were set to zero.

This constraint means that any movement in any direction on these surfaces is completely restricted. Thus, the model is rigidly fixed along the two end regions and the basal surface of the mandible, ensuring that no horizontal (X, Y) or vertical (Z) displacement occurs in these regions during the analysis. This approach preserves the anatomical integrity of the mandible in the simulation, allowing the loads to generate stress and deformation only in the regions of interest. In addition, the rigidity provided by the fixation at the ends and base prevents the model from undergoing rigid body motion in the numerical solution, enabling the attainment of stable and physically meaningful results.

Under these loading and boundary conditions, a total of ten linear static and twenty nonlinear static simulations were performed across five distinct mandibular models. This comprehensive simulation strategy provides a more robust representation of implant biomechanics compared to most FEA studies, which typically use only one or two simplified loading cases. This broader approach helps identify stress variations under a range of clinically relevant conditions and enhances the validity of the conclusions.

The applied forces were distributed evenly across 10 nodes located on the occlusal surface of the crown. This approach ensured consistent load application and improved simulation stability.

### 2.6. Quantitative Model Data

To support the validity and mesh convergence of the finite element models, the number of nodes and elements in each analysis configuration is summarized in Table 2. Mesh refinement was carefully performed within a range of 0.1–0.25 mm tria elements on the surface and tetrahedral meshing in the volume to ensure accuracy in stress computation. All models were checked for mesh quality and convergence.

The nodal and element counts for the five models are listed in Table 2.

Compared to typical finite element studies in dental biomechanics—which often include only one or two simplified model geometries [9,12]—this study employed five distinct anatomical models and 30 total simulations (10 linear + 20 nonlinear). This comprehensive strategy provides greater biomechanical insight and clinical relevance by accommodating anatomical variability and loading scenarios.

### 2.7. Assembly and Contact Definitions Between Components

For accurate analysis results, interfacial contact definitions between model components were established in the analysis software.

The implant–bone interface conditions were adjusted according to the loading protocol:Immediate Loading: The friction coefficient at the interface was set to 0.3, simulating incomplete osseointegration.Early Loading: Partial osseointegration was assumed, and the friction coefficient was set to 0.5.Delayed Loading: Complete osseointegration was assumed, and a “FREEZE” contact condition was applied, implying no relative motion between components.

Other contacting components were also defined using the “FREEZE” contact model, assuming full coupling between the surfaces during loading.

For accurate and clinically relevant results in finite element analysis (FEA), interfacial contact definitions between all structural components were explicitly modeled within the ALTAIR OptiStruct solver.

To simulate different stages of osseointegration, the implant–bone interface was defined with varying contact properties based on the applied loading protocol:Immediate Loading: A frictional contact with a coefficient of μ = 0.3 was defined to represent a clinical condition where osseointegration has not yet occurred. This allows for limited relative micromotion between the implant and the bone, mimicking the initial healing period.Early Loading: A frictional contact with a higher friction coefficient of μ = 0.5 was used to simulate partial osseointegration and increased mechanical interlocking, reflecting conditions after initial bone healing but before full integration.Delayed Loading: For fully osseointegrated implants, a “FREEZE” contact condition was applied. This model enforces complete coupling between surfaces, preventing any relative displacement or sliding, thereby simulating mature bone–implant integration.

These friction values and boundary definitions are consistent with established FEA studies investigating bone–implant biomechanics [9,12,14].

All remaining component interfaces were defined using the “FREEZE” contact model, assuming full rigid connection during function, including the following:Crown–abutment;Abutment–implant;Screw–implant;Implant–crown.

This assumption is based on clinical protocols where torque-controlled fastening and passive-fit restorations are used, minimizing micromovements and stress concentration between components.

This comprehensive contact definition strategy enhances the biomechanical validity of the simulation, enabling accurate prediction of stress transmission patterns under both linear and nonlinear conditions.

## 3. Results

In this study, vertical (0°) and oblique (30°) forces were applied and compared across three loading scenarios (immediate, early, and delayed) using implants 6 mm and 12 mm in length.

The von Mises stress values presented in the tables correspond to local peak stresses rather than averaged values. For cortical and trabecular bone, these values were measured at two anatomically relevant reference points: the cervical (P1) and apical (P3) regions adjacent to the implant body. In the implant and abutment, stress data were collected from internal locations with the highest concentration of mechanical stress. For the mandibular canal, measurements were taken from the volume immediately surrounding the canal walls.

First, the von Mises stresses occurring in the implant, abutment, and mandibular canal regions, as well as the stress distribution in cortical and trabecular bone tissues, were evaluated under vertical loading applied to the normal mandible model (Table 3).

Subsequently, the same model was subjected to oblique loading, and the resulting stress values in these regions were compared (Table 4).

The von Mises stress values presented in Table 3 and Table 4 represent peak values measured at specific regions of interest. In cortical and trabecular bone, these were taken from the cervical (P1) and apical (P3) points adjacent to the implant body. The reported values correspond to local maximum stress concentrations, which are critical for evaluating biomechanical performance under different loading conditions.

In the atrophic mandibular model, only the 6 mm implant was evaluated. Stress distribution patterns were examined under three loading protocols (immediate, early, and delayed) using both vertical and oblique loading conditions. The stress responses in the peri-implant bone tissues are summarized in Table 5 and Table 6.

The von Mises stress values presented in the tables reflect the peak stress concentrations measured in specific regions of interest. In cortical and trabecular bone, values were obtained at two representative points: the P1 (cervical) and P3 (apical) regions surrounding the implant. These localized measurements allow for evaluating stress transmission near critical implant interfaces, particularly in compromised bone quality.

As seen in Table 5 and Table 6, the maximum compressive stress observed under oblique loading in the cortical bone (−227.051 MPa) approaches the reported compressive strength limit of cortical bone (approximately 130–230 MPa). This may indicate a potential risk of biomechanical overloading in atrophic mandibular conditions, particularly under immediate loading protocols. These findings underscore the importance of careful case selection and appropriate loading timing in clinical practice to reduce the likelihood of excessive stress and possible bone failure.

Finally, in the grafted atrophic mandible model, von Mises stress values and peri-implant stress distributions in cortical and trabecular bone were obtained for both 6 mm and 12 mm implants under vertical and oblique loading (Table 7 and Table 8).

As seen in Table 7 and Table 8, the grafted atrophic mandibular model exhibited lower von Mises stress values in cortical and trabecular bone compared to the atrophic model without graft support. The highest compressive stress observed in the cortical bone under oblique loading (−38.615 MPa) was well below the critical failure threshold of cortical bone (approximately −130 to −230 MPa). This indicates that grafting improved biomechanical performance by redistributing stress and reducing concentration zones around the implant. Additionally, longer implants (12 mm) generally resulted in lower stress magnitudes, particularly in the apical region (P3), suggesting better load transfer to deeper trabecular bone. These findings emphasize the clinical relevance of bone grafting in compromised mandibles and support the use of delayed loading protocols to minimize overload risk during healing.

### Parameter-Based Comparisons

#### 3.1.1. Von Mises Stresses in the Implant Body

Across all models and scenarios, the von Mises stresses observed in the implant body varied significantly depending on the loading type and implant length. In the normal mandible model under vertical loading, stresses ranged from 104.161 MPa to 105.237 MPa for the 6 mm implant, whereas the 12 mm implant exhibited values between 118.273 MPa and 118.801 MPa, indicating that the longer implant generated approximately 13–14% higher stress (Figure 10). The effect of loading time in this model was limited, showing only minor percentage changes between immediate and delayed loading.

Under oblique loading, however, the stresses in the implant body increased by approximately 3.5 times compared to under vertical loading. Stress values ranged from 346.366 MPa to 365.352 MPa for the 6 mm implant and from 452.020 MPa to 465.698 MPa for the 12 mm implant. Although a slight reduction was observed in delayed loading, the overall stress levels remained high. This finding clearly demonstrates the moment effect generated by inclined forces and how implant length amplifies this biomechanical load.

In the atrophic mandible model, only the 6 mm implant was analyzed. Stresses remained relatively constant in the range of 110.299 MPa to 111.314 MPa under vertical loading (Figure 11), while significantly higher values between 387.755 MPa and 403.720 MPa were recorded under oblique loading (Figure 12). A decrease of approximately 16 MPa was observed in delayed loading. These findings suggest that atrophic bone is less sensitive to loading time but is strongly affected by oblique forces, which increase the stress by a factor of three to four.

In the grafted atrophic mandible model, both implant lengths were evaluated. Under vertical loading, stress values in the implant body ranged from 104.278 MPa to 105.625 MPa for the 6 mm implant, and from 118.435 MPa to 128.247 MPa for the 12 mm implant (Figure 13). Under oblique loading, stress levels were measured between 389.284 MPa and 392.595 MPa for the 6 mm implant, and between 470.746 MPa and 471.828 MPa for the 12 mm implant (Figure 14). A limited reduction in stress was observed in delayed loading for both lengths; however, even with graft augmentation, higher stress levels persisted as implant length increased.

The reported von Mises stress values represent peak values observed at high-stress concentration zones within the implant body.

Considering all conditions, the lowest von Mises stress was recorded in the normal mandible model under vertical loading with the 6 mm implant in the immediate loading scenario, measured at 104.161 MPa (Figure 15). The highest stress was observed in the grafted atrophic model under oblique loading with the 12 mm implant in the immediate loading scenario, measured at 471.828 MPa (Figure 16).

Loading type and implant length were identified as the most influential parameters affecting stress distribution in the implant body. Although graft application partially reduced excessive stress accumulation in atrophic mandibles, its effectiveness was limited, especially under oblique forces and with longer implants.

#### 3.1.2. Von Mises Stresses at the Abutment Level

von Mises stresses observed at the abutment level varied depending on mandibular morphology, implant length, and loading conditions.

In the normal mandible model under vertical loading, stress values for the 6 mm implant were recorded as 124.073 MPa (immediate), 124.096 MPa (early), and 124.594 MPa (delayed). For the 12 mm implant, the corresponding values were 127.291 MPa, 127.544 MPa, and 128.319 MPa, respectively. The increase in implant length resulted in an additional stress of approximately 3–4 MPa at the abutment level. The timing of loading had no meaningful effect under vertical loading.

When oblique loading was applied, abutment stresses increased by approximately 3.5 times compared to vertical loading. For the 6 mm implant, the stresses were 441.488 MPa (immediate), 442.256 MPa (early), and 442.839 MPa (delayed) (Figure 17).

For the 12 mm implant, values reached 458.783 MPa, 459.062 MPa, and 459.678 MPa, respectively (Figure 18). The longer implant generated an additional 17–18 MPa of stress under oblique loading, yet the influence of loading timing remained negligible.

In the atrophic mandible model, only the 6 mm implant was assessed. Under vertical loading, abutment stresses were measured as 132.491 MPa (immediate), 132.856 MPa (early), and 134.434 MPa (delayed), showing a minimal increase over time (Figure 19). In contrast, oblique loading significantly elevated the stresses to 463.917 MPa, 465.360 MPa, and 466.400 MPa, confirming the influence of angled force application (Figure 20).

In the grafted atrophic mandible model, both 6 mm and 12 mm implants were evaluated. Under vertical loading, stresses ranged from 127.166 MPa to 127.189 MPa for the 6 mm implant, and from 127.359 MPa to 128.640 MPa for the 12 mm implant. Under oblique loading, stress levels ranged from 447.861 MPa to 449.454 MPa (6 mm implant), and from 458.023 MPa to 459.382 MPa (12 mm implant). The use of a longer implant once again resulted in an increase of approximately 10–12 MPa at the abutment level.

The von Mises stress values presented here represent peak stresses extracted from the abutment region. These values correspond to maximum localized stress concentrations, rather than averaged distributions.

The lowest von Mises stress at the abutment level was observed in the normal mandible model under vertical loading (0°) using a 6 mm implant during immediate loading (124.073 MPa) (Figure 21).

The highest abutment stress occurred in the atrophic mandible model under oblique loading (30°) with a 6 mm implant during delayed loading (466.400 MPa) (Figure 22).

These findings indicate that the type of loading (vertical vs. oblique) is the most decisive parameter influencing stress concentration in the abutment region. Implant length has a secondary but consistent effect, while loading timing showed no clinically relevant impact.

#### 3.1.3. Von Mises Stresses Around the Mandibular Canal

von Mises stresses around the mandibular canal varied according to mandibular morphology, implant length, loading direction, and loading protocol.

In the normal mandible model under vertical loading, stresses ranged from 0.937 to 0.865 MPa for the 6 mm implant, and from 1.827 to 1.765 MPa for the 12 mm implant. Doubling the implant length nearly doubled the stress level in the surrounding canal region. Under oblique loading, stress values ranged from 0.768 to 0.603 MPa (6 mm implant) and from 1.133 to 1.065 MPa (12 mm implant). All measured stresses in this model remained below the generally accepted 2 MPa threshold for bone adaptation, indicating a low-risk biomechanical environment.

In the atrophic mandible model, only the 6 mm implant was evaluated. Under vertical loading, canal stresses ranged from 3.656 to 3.146 MPa (Figure 23). Under oblique loading, they ranged from 5.227 to 3.797 MPa (Figure 24). The maximum value of 5.227 MPa was observed under oblique immediate loading. Although higher than in other groups, this value still remained below the reported 6–10 MPa threshold associated with potential inferior alveolar nerve damage in clinical studies.

In the grafted atrophic mandible model, stress values under vertical loading ranged from 1.891 to 1.888 MPa (6 mm implant) and from 2.480 to 2.287 MPa (12 mm implant). Under oblique loading, values ranged from 2.171 to 2.051 MPa for the 6 mm implant and from 2.971 to 2.263 MPa for the 12 mm implant. The use of grafts effectively reduced stress values compared to the atrophic model. However, increasing the implant length once again introduced a moderate increase in canal stress.

The lowest stress around the mandibular canal (0.603 MPa) was recorded in the normal mandible model under oblique loading with a 6 mm implant during delayed loading (Figure 25).

The highest stress (5.227 MPa) occurred in the atrophic mandible model under oblique loading with a 6 mm implant during immediate loading (Figure 26).

These results confirm that, in all configurations, von Mises stresses around the mandibular canal remained below critical neurovascular damage thresholds. Therefore, all evaluated implant protocols demonstrated a biomechanically safe stress profile with respect to the mandibular canal.

#### 3.1.4. Maximum (Tensile) Principal Stresses at Cortical Bone Point P1

The maximum principal tensile stresses at the P1 point, located in the cervical region of the cortical bone near the implant neck, demonstrated significant variation based on mandibular morphology, loading direction, implant length, and loading timing.

In the normal mandible model under vertical loading, tensile stresses for the 6 mm implant were 28.335 MPa (immediate), 28.094 MPa (early), and 22.956 MPa (delayed). With the 12 mm implant, values decreased to 17.949 MPa, 16.581 MPa, and 15.519 MPa, respectively. This indicates that increasing the implant length led to a 35–40% reduction in tensile stress, and delayed loading was associated with further stress attenuation.

Under oblique loading, stresses increased significantly. For the 6 mm implant, values reached 72.730 MPa (immediate), 70.227 MPa (early), and 51.201 MPa (delayed), while the 12 mm implant showed lower but persistent levels: 44.052 MPa, 44.039 MPa, and 43.731 MPa, respectively. These findings confirm that angled forces nearly tripled the stress, although a reduction was still observed over time.

In the atrophic mandible model, only 6 mm implants were analyzed. Vertical loading resulted in tensile stresses of 84.186 MPa (immediate), 79.110 MPa (early), and 60.650 MPa (delayed) (Figure 27). Under oblique loading, the stresses escalated to 179.798 MPa (immediate), 157.496 MPa (early), and 118.069 MPa (delayed) (Figure 28).

This value (179.798 MPa) was the highest recorded cortical tensile stress in the entire study and approached the reported tensile strength limit of cortical bone (~130–200 MPa) (Figure 29). Such elevated stress levels under immediate oblique loading in atrophic conditions may suggest a potential risk of microfracture or marginal bone loss.

In the grafted atrophic mandible, vertical loading resulted in lower stresses: 16.391 MPa (immediate and early) and 15.221 MPa (delayed) for the 6 mm implant, and 16.828 MPa, 16.675 MPa, and 15.485 MPa for the 12 mm implant. These results suggest that grafting significantly reduced cervical tensile stress.

Under oblique loading in the grafted model, stress values for the 6 mm implant were 15.832 MPa (immediate), 14.728 MPa (early), and 13.429 MPa (delayed), while the 12 mm implant exhibited 32.591 MPa, 25.856 MPa, and 21.432 MPa. The lowest P1 tensile stress, 13.429 MPa, was recorded with the 6 mm implant under delayed oblique loading (Figure 30).

These findings underscore that graft application and delayed loading protocols are effective in reducing cervical cortical stress, thereby potentially improving implant success. However, the use of longer implants under oblique loading may counteract the benefits by reintroducing higher tensile forces at the crestal bone interface (Figure 31).

#### 3.1.5. Maximum (Compressive) Principal Stresses at Point P3 in Cortical Bone

Maximum compressive stresses measured at the P3 point (apical region of the cortical bone) varied significantly depending on bone morphology, loading direction, implant length, and loading time.

In the normal mandible under vertical loading, compressive stresses for the 6 mm implant were −28.180 MPa (immediate), −27.206 MPa (early), and −25.275 MPa (delayed); for the 12 mm implant, the values were −21.025 MPa, −20.645 MPa, and −20.550 MPa, respectively. These findings indicate that increasing the implant length reduced apical compressive stress by approximately 25%, while loading time resulted in a progressive stress decrease.

Under oblique loading, compressive stresses increased markedly. For the 6 mm implant, stress values were −87.935 MPa (immediate), −85.808 MPa (early), and −56.861 MPa (delayed); for the 12 mm implant, they were −52.519 MPa, −52.447 MPa, and −44.907 MPa. These results demonstrate that oblique forces increased compressive stress by approximately two to three times, although this effect was mitigated with delayed loading.

In the atrophic mandible model, only the 6 mm implant was assessed. Under vertical loading, compressive stresses were −101.279 MPa (immediate), −90.546 MPa (early), and −71.909 MPa (delayed) (Figure 32). Under oblique loading, stresses reached the highest levels across all simulations: −227.051 MPa (immediate), −225.500 MPa (early), and −131.396 MPa (delayed) (Figure 33). These values approach the upper limits of cortical bone compressive strength, which ranges between 170 and 200 MPa, suggesting a potential risk for localized cortical microdamage under oblique forces in severely resorbed bone conditions.

In the grafted atrophic mandible model, graft augmentation markedly reduced compressive stresses in the apical cortical region. Under vertical loading, stresses ranged between −14.471 MPa and −13.395 MPa for the 6 mm implant, and from −25.131 MPa to −23.007 MPa for the 12 mm implant. Under oblique loading, compressive stress values were between −38.615 MPa and −21.010 MPa (6 mm implant), and between −17.908 MPa and −15.442 MPa (12 mm implant). These findings confirm that grafting effectively reduces compressive loading, particularly under vertical force applications. However, stress levels still rise with longer implants and oblique forces.

The lowest compressive stress was −13.395 MPa, observed in the grafted atrophic mandible with a 6 mm implant under vertical delayed loading (Figure 34). The highest compressive stress was −227.051 MPa, recorded in the atrophic mandible under oblique immediate loading with a 6 mm implant (Figure 35).

All values remained within the accepted range for cortical bone compressive strength, except for one condition, where the peak (−227.051 MPa) slightly exceeded reported thresholds. This suggests that, in atrophic bone without graft support, oblique immediate loading may pose a biomechanical risk, particularly in the apical cortical region. In contrast, graft application appears to effectively restore structural integrity and maintain compressive stresses within clinically safe limits across all conditions.

#### 3.1.6. Maximum (Tensile) Principal Stresses at the P1 Point in Trabecular Bone

The maximum tensile stresses at the P1 point—representing a cervical region within the trabecular bone—varied depending on mandibular structure, implant length, loading direction, and timing.

In the normal mandible model under vertical loading, the 6 mm implant generated stress values of 0.930 MPa (immediate), 0.882 MPa (early), and 0.870 MPa (delayed) (Figure 36). With the 12 mm implant, these values dropped to 0.551 MPa, 0.565 MPa, and 0.475 MPa, respectively (Figure 37). Increasing implant length reduced tensile stresses by approximately 35–50% in this region.

Under oblique loading, the stresses increased considerably. For the 6 mm implant, values were 4.792 MPa (immediate), 4.135 MPa (early), and 4.123 MPa (delayed); for the 12 mm implant, the results were 1.852 MPa, 1.653 MPa, and 1.549 MPa. Angled forces induced a three-to-four-fold increase in stress; however, this effect was slightly mitigated with delayed loading. The highest value under this condition—4.792 MPa—was recorded with the 6 mm implant during immediate loading.

In the atrophic mandible model, only the 6 mm implant was assessed. Under vertical loading, the values were 1.581 MPa (immediate), 1.467 MPa (early), and 1.411 MPa (delayed). Under oblique loading, the tensile stresses increased to 5.261 MPa (immediate), 5.257 MPa (early), and 5.069 MPa (delayed). The highest tensile stress in all simulations —5.261 MPa—was recorded under oblique immediate loading (Figure 38).

In the grafted atrophic mandible model, vertical loading with a 6 mm implant resulted in stress values of 0.409 MPa (immediate), 0.405 MPa (early), and 0.403 MPa (delayed). For the 12 mm implant, values were 0.634 MPa, 0.567 MPa, and 0.514 MPa. Under oblique loading, tensile stresses were 0.591 MPa, 0.572 MPa, and 0.539 MPa for the 6 mm implant, and 1.115 MPa, 1.081 MPa, and 0.977 MPa for the 12 mm implant. The lowest tensile stress, 0.403 MPa, was observed in this group during vertical delayed loading with a 6 mm implant (Figure 39).

Overall, tensile stresses at the P1 point in trabecular bone ranged from 0.403 MPa to 5.261 MPa, remaining well below the ultimate tensile strength of trabecular bone (~10–15 MPa). These results indicate that, regardless of the loading condition or implant configuration, the biomechanical integrity of the trabecular region was preserved. However, oblique forces in atrophic bone without graft support represent the most vulnerable scenario and require clinical caution.

#### 3.1.7. Minimum (Compressive) Principal Stresses at P3 Point in Trabecular Bone

The minimum (compressive) principal stresses at point P3, located in the apical trabecular bone region, showed substantial variation depending on mandibular morphology, implant length, loading direction, and loading protocol.

In the normal mandible model under vertical loading, compressive stress values for the 6 mm implant were −1.744 MPa (immediate), −1.625 MPa (early), and −0.675 MPa (delayed). For the 12 mm implant, the corresponding values were −1.254 MPa, −1.251 MPa, and −0.525 MPa. These results indicate that increasing the implant length led to a reduction in compressive stress of approximately 30–60%. Additionally, a gradual decrease in stress was observed over time with delayed loading (Figure 40).

Under oblique loading, compressive stresses increased significantly. For the 6 mm implant, stresses were −12.429 MPa (immediate), −11.162 MPa (early), and −3.777 MPa (delayed). For the 12 mm implant, the values were −10.430 MPa, −9.747 MPa, and −2.705 MPa. These findings demonstrate that inclined forces elevated compressive stress up to tenfold compared to vertical loading. However, this increase was partially mitigated with delayed loading (Figure 41). The highest compressive stress observed in the normal mandible model was −12.429 MPa, recorded with the 6 mm implant under immediate oblique loading.

In the atrophic mandible model, only the 6 mm implant was evaluated. Under vertical loading, the compressive stresses were −2.328 MPa (immediate), −1.824 MPa (early), and −1.635 MPa (delayed). Under oblique loading, the values were considerably higher, reaching −19.102 MPa (immediate), −17.006 MPa (early), and −6.797 MPa (delayed). The value of −19.102 MPa represented the highest compressive stress recorded across all trabecular scenarios (Figure 42).

In the grafted atrophic mandible model, compressive stresses were notably reduced for both implant lengths. Under vertical loading, the 6 mm implant produced stresses of −0.786 MPa, −0.780 MPa, and −0.745 MPa (immediate to delayed), while the 12 mm implant yielded −1.508 MPa, −1.500 MPa, and −1.484 MPa. Under oblique loading, compressive stresses for the 6 mm implant were −0.869 MPa, −0.851 MPa, and −0.810 MPa, and, for the 12 mm implant, −6.715 MPa, −5.548 MPa, and −5.138 MPa. The lowest value across all conditions (−0.745 MPa) was observed with the 6 mm implant under delayed vertical loading (Figure 43).

In summary, trabecular compressive stresses ranged from −0.745 MPa to −19.102 MPa across all models and conditions. Although higher values were recorded in the atrophic mandible under oblique loading, all stresses remained below the critical compressive strength threshold for trabecular bone. These findings suggest that graft application enhances biomechanical safety by effectively reducing compressive stress in the apical trabecular region under both vertical and oblique loading.

## 4. Discussion

In this study, the effects of mandibular morphology (normal, atrophic, and grafted), implant length (6 mm and 12 mm), and loading timing (immediate, early, and delayed) on peri-implant stress distribution were evaluated three-dimensionally using advanced finite element analysis (FEA). Vertical (200 N) and oblique (100 N, 30°) loading scenarios were applied to anatomical models reconstructed from high-resolution CT data, and von Mises stress, maximum tensile stress (P1), minimum compressive stress (P3) in trabecular bone, and the stress profile around the mandibular canal were analyzed.

The scientific hypotheses were structured according to this parametric design. The null hypothesis—that implant biomechanics function independently of bone morphology—was rejected. Stress concentrations were significantly higher in atrophic models, while graft-supported bone exhibited more physiological stress distribution. Similarly, implant length clearly affected stress transmission; short implants caused higher stress concentrations in the surrounding bone, while longer implants facilitated a wider distribution of forces, thus reducing marginal stress. Graft material provided both volumetric and elastic support to the bone, allowing for a more balanced mechanical environment. Additionally, synchronizing the loading time with the progress of osseointegration produced more favorable biomechanical outcomes; delayed loading scenarios resulted in lower and more uniformly distributed stresses in both cortical and trabecular bone.

The findings confirmed that bone morphology plays a key role in determining peri-implant stress behavior. In the normal mandible model, stress was homogeneously distributed and remained within physiological limits. In contrast, atrophic models— particularly under oblique loading—exhibited elevated stress levels at both P1 and P3 points. Graft application successfully mitigated these stress concentrations and promoted a more balanced load distribution [22,23,24,25,26].

Implant length demonstrated a dual biomechanical effect. While longer implants promoted more efficient stress dispersion within both cortical and trabecular bone—thereby lowering peri-implant tensile and compressive stress levels—they concurrently induced higher von Mises stress concentrations at the implant–abutment interface. This internal stress increase may elevate the risk of mechanical complications, such as screw loosening, abutment fracture, or implant component fatigue, particularly under oblique loading conditions [27,28,29].

Loading timing emerged as another critical biomechanical factor. Immediate loading led to higher tensile stress in cortical bone and greater compressive stress in trabecular bone, whereas delayed loading lowered these values and resulted in more homogeneous stress distribution. This finding supports the notion that, as osseointegration advances, tissue tolerance improves and load transmission becomes more physiological [30,31,32,33].

Implant length also directly influenced stress transfer in both peri-implant bone and prosthetic components. Short implants induced more localized stress accumulation in cortical and trabecular bone, while longer implants distributed loads more broadly, reducing peak stresses. Specifically, 12 mm implants provided reductions of 30–60% in P1 and P3 stress values. However, longer implants also caused an increase in von Mises stress at the abutment interface, which could predispose the system to mechanical complications such as screw loosening or abutment fracture [9,27,29,34].

In particular, abutment stress levels reached values as high as 466 MPa under oblique loading, clearly exceeding the fatigue thresholds reported in the literature [35]. For example, finite element analysis has shown that von Mises stresses in custom abutments range from 229 to 302 MPa—and that 94.4% of designs exhibit fatigue limits at or above those stresses—suggesting that the 466 MPa level likely surpasses fatigue endurance levels [35]. Furthermore, under ISO 14801–based oblique loading (100 N), increased abutment screw torque significantly elevated residual stresses and consequently reduced the fatigue life of the screw [36]. Such elevated stresses may predispose the system to mechanical complications—including abutment screw loosening, deformation, or fracture—especially under cyclic functional loads [36]. Therefore, these findings highlight the critical need for prosthetic designs that minimize lateral forces and optimize load direction to reduce mechanical stress on the implant–abutment interface.

Grafted models effectively reduced cortical stress intensity by up to 28% and allowed for more favorable load distribution in the trabecular region. The elasticity and volume of graft material supported stress transfer within physiological thresholds, improving implant stability [28,37,38,39]. However, in scenarios involving insufficient graft volume or incomplete maturation—particularly when combined with long implants—localized compressive stress may increase, potentially leading to delayed resorption [40,41,42].

Bone density was also a major determinant of stress behavior. The normal mandible (D2 bone type) absorbed stress efficiently and maintained maximum values within biomechanical safety margins. In contrast, the atrophic mandible (D4 bone type) exhibited elevated stress concentrations under oblique loading, heightening the risk of micromotion and subsequent resorption [43]. Following grafting, the simulated D3 bone condition offered a more balanced stress profile and favorable support for sustained osseointegration.

Implant surface topography further contributed to stress dynamics. The SLA-treated Straumann implants employed in this study demonstrated strong primary stability, facilitated early osseointegration, and maintained von Mises stress within acceptable biological levels, particularly in delayed loading conditions [44,45]. Variations in surface roughness and friction coefficients affected micromechanical interactions, enabling the finite element model to simulate stress behavior on both macro- and micro-scales [46,47].

The osseointegration process is a dynamic biological progression that shapes stress transfer over time. In the present model, loading protocols (immediate, early, delayed) were simulated using variable friction coefficients (0.3, 0.5, and freeze contact), thereby enabling a parametric assessment of biomechanical outcomes across different healing phases. Immediate loading conditions produced elevated tensile stress in cortical regions and compressive stress in trabecular areas, while delayed loading significantly attenuated these values, indicating enhanced mechanical integration and predictability [30,31,32,33].

These friction coefficient values were selected based on previous FEA studies that modeled osseointegration stages using similar parameters to reflect varying bone–implant contact levels [9,13,15].

Specifically, 0.3 reflects the limited contact typical of immediate loading, 0.5 simulates early mechanical engagement, and ‘freeze’ represents fully integrated conditions approximating ideal osseointegration. This approach enables approximation of progressive biomechanical changes during healing while maintaining computational tractability.

According to the biomechanical thresholds established in the literature, stress levels around 2 MPa stimulate bone adaptation, while values exceeding 4 MPa may trigger bone resorption. For the inferior alveolar nerve, 6–10 MPa is cited as the critical threshold for neurosensory disturbance [48,49,50]. In this study, von Mises stress levels surrounding the mandibular canal were 3–4 times higher in the atrophic mandible compared to normal and grafted conditions. However, all values remained below reported neurophysiological damage limits [51,52]. Despite this, adherence to the “≥2 mm distance from the canal” rule remains essential in implant placement to minimize risk [53,54,55].

Moreover, this study confirmed that marginal bone loss around implants is closely associated with cortical stress accumulation. Bone loss of up to 1.5 mm during the first year is considered physiological, whereas greater loss may indicate underlying pathology [56]. In grafted models, delayed loading appeared to mitigate this risk [21]. In trabecular areas, the combination of short implants and oblique loading increased localized stress and the potential for micromotion, which could negatively affect osseointegration [57,58].

Oblique loading consistently induced higher stress levels compared to vertical loading across all models—particularly manifesting as increased compressive stress in trabecular bone. These results suggest that oblique force vectors elevate the risk of microdamage at the cortical–trabecular interface and may adversely affect implant prognosis [59]. Consequently, clinical protocols should prioritize occlusal adjustments that minimize oblique force components, direct loads axially, and ensure controlled stress transmission, especially in low-density bone areas [60,61,62].

Implant design parameters—such as diameter, thread geometry, platform switching, and surface morphology—also exert a significant influence on peri-implant stress distribution. The SLA-coated Straumann implant model used in this study, with its medium diameter (4.1 mm), demonstrated favorable primary stability attributable to its surface topography. According to the literature, wider implants reduce stress accumulation by increasing the contact area with bone, while narrower implants may amplify peripheral stress concentrations, particularly in low-density bone environments [63,64]. Furthermore, micro-rough surfaces such as SLA and SLActive enhance osteoblast adhesion and activity, thereby promoting early osseointegration and providing notable biomechanical advantages [44,45,46,47,65,66].

However, while longer implants improve the uniformity of stress distribution in both cortical and trabecular bone, they tend to increase von Mises stress at the implant– abutment interface. This biomechanical load concentration may lead to mechanical complications such as screw loosening, abutment fractures, or deformation of the implant body—complications more frequently associated with extended implant lengths [9,27,29,34]. Therefore, prosthetic planning should account not only for bone volume and density but also for mechanical factors related to stress transmission within the implant system.

Peri-implantitis is influenced not only by microbial factors but also by biomechanical stress patterns. Particularly under excessive loading conditions, the accumulation of stress in peri-implant cortical and trabecular bone can initiate pathological responses in both soft and hard tissues [67]. Elevated mechanical stress may trigger osteoclastic activity, creating a predisposition for marginal bone loss. This risk is heightened in clinical cases involving reduced tissue tolerance, such as atrophic or grafted bone regions. In such high-risk scenarios, patient-specific biomechanical assessments are crucial during the treatment planning phase to preserve peri-implant tissues and optimize long-term implant success.

Integrating FEA-based biomechanical analysis into clinical decision-making facilitates patient-specific optimization of critical variables such as implant length, diameter, loading timing, grafting requirements, and bone quality. Recent advancements in artificial intelligence have made it possible to analyze such multi-parametric datasets, thereby improving the predictability of complication risks and supporting evidence-based planning [68,69,70]. This underscores the growing relevance of personalized biomechanical treatment strategies in implantology and represents a tangible advancement toward improving long-term clinical outcomes.

One limitation of this study is the absence of a formal mesh convergence analysis. However, the selected element size range (0.1–0.25 mm) was adopted based on multiple validated studies involving comparable mandibular implant models. These studies have consistently demonstrated that this range provides an optimal balance between computational efficiency and solution accuracy.

The findings of this study emphasize that peri-implant biomechanical evaluations are inherently multidimensional. When variables such as bone density, implant geometry, graft volume, the loading vector, and functional timing are collectively considered, clinicians can formulate optimized surgical and prosthetic strategies. In high-risk cases—such as atrophic mandibles or compromised bone quality—digital decision-support systems that incorporate these variables may help prevent both mechanical complications and biological failures.

However, several limitations of the current study must be acknowledged. First, to maintain computational feasibility, simplifications were made in anatomical geometry and material properties. All models utilized homogenized material definitions and assumed linear-elastic behavior, which may not fully capture the nonlinear and anisotropic characteristics of bone remodeling or the mechanical behavior at the implant–bone interface under prolonged functional loading. Second, the applied loading conditions were static rather than dynamic, which does not fully reflect the time- dependent nature of in vivo masticatory forces. Third, although the models were constructed using high-resolution CT data, the simulations were not patient-specific. Therefore, the generalizability of the results to all clinical scenarios should be interpreted with caution.

In terms of visualization, standardized representative images were provided for each model group in the Materials and Methods section. Since all models shared identical meshing density, boundary conditions, and loading parameters, redundant visuals were excluded, and only illustrative figures were included for clarity.

Furthermore, only a single oblique force direction (30°) was applied during the simulations. This angle was functionally selected to represent a clinically relevant occlusal vector on the buccal cusp of the mandibular first molar (functional tuberculum). Modeling additional loading angles would have substantially increased the number of models and computational time, which could have reduced the clarity and interpretability of the results. Therefore, this simplification should be taken into account when evaluating the findings.

Additionally, the assumption of isotropic, homogeneous, and linear-elastic material properties may not fully capture the anisotropic and porous nature of graft materials such as Bio-Oss, nor the distinct mechanical behavior between cortical and trabecular bone regions.

Despite these limitations, the results of this study underscore the importance of personalized biomechanical planning in implantology. Regardless of whether short implants are used in atrophic bone or graft-supported long implants are placed, anatomical, biological, and mechanical parameters must be holistically evaluated. Not only implant length and surface characteristics, but also loading direction, graft maturation status, osseointegration stage, and prosthetic design contribute significantly to long-term success. For example, if immediate loading is planned, elevated cortical stress should be anticipated. When oblique forces cannot be avoided—particularly in the posterior mandible—prosthetic design must prioritize centralized occlusal contacts. In atrophic regions close to the mandibular canal, even stress values below neurosensory thresholds may reduce safety margins. Therefore, finite element analysis offers a valuable tool for integrating these factors into a patient-specific decision-making process, especially in high-risk cases.

## 5. Conclusions

Implant length significantly influenced peri-implant stress distribution.

Compared to 6 mm implants, 12 mm implants reduced the maximum principal stress in trabecular bone by approximately 40–50%.

For instance, in the normal mandible under oblique loading, stress at the trabecular P1 point decreased from 4.792 MPa (6 mm) to 1.852 MPa (12 mm).

Loading timing played a critical role in stress modulation.

Immediate loading led to elevated cortical stress due to incomplete osseointegration.

Delayed loading with the same implant length significantly reduced cortical stress and promoted more stable load transfer.

Oblique loading was the most potent factor in amplifying stress within the implant system.

Under identical conditions, oblique loading increased von Mises stress by 3–4 times compared to vertical loading.

In a 6 mm implant within the normal mandible, von Mises stress rose from 104.161 MPa (vertical) to 365.352 MPa (oblique).

Mandibular bone condition markedly affected stress patterns.

In the atrophic mandible, compressive stress peaked at −227.051 MPa, while, in the grafted model, this dropped to −38.615 MPa, indicating that grafts enhance cortical load-bearing capacity.

The mandibular canal region proved highly sensitive to loading conditions.

In the atrophic mandible with a 6 mm implant and oblique loading, canal stress reached 5.227 MPa, nearing the 6 MPa threshold for nerve damage.

Although not exceeded, the reduced safety margin necessitates cautious implant positioning.

Grafting effectively reduced stress in both cortical and trabecular bone.

In the trabecular P3 region of the grafted atrophic mandible, the compressive stress was –0.786 MPa, compared to −2.328 MPa in the non-grafted model.

This confirms grafting’s dual role in volume augmentation and stress absorption.

Stress distribution varied between the implant’s cervical and body regions.

In delayed loading conditions, stress increased in the cervical region but decreased in the implant body.

This suggests that, as osseointegration progresses, stress is gradually transferred toward the cervical zone, indicating a shift in the mechanical load path over time.

## Figures and Tables

**Figure 1 bioengineering-12-00888-f001:**
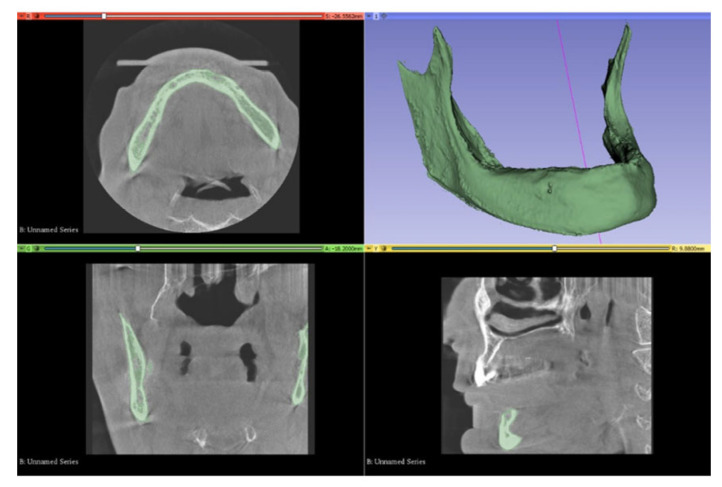
Three-dimensional reconstruction of the mandible and associated axial, coronal, and sagittal CBCT slices following segmentation and region-of-interest (ROI) isolation. (The green-highlighted regions represent the segmented mandibular bone).

**Figure 2 bioengineering-12-00888-f002:**
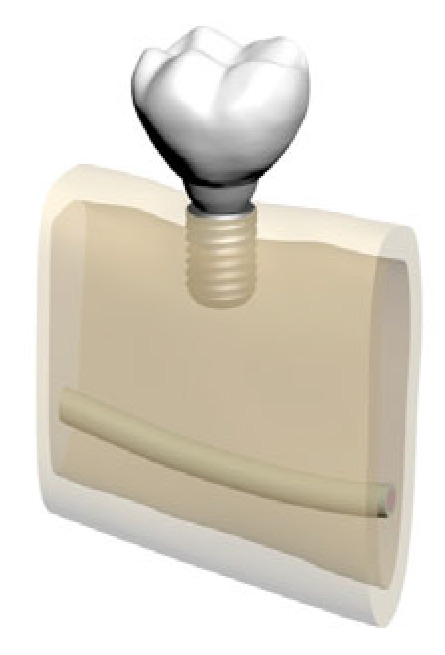
Model 01: Normal mandible (D2), 10 mm bone width, 6 mm implant.

**Figure 3 bioengineering-12-00888-f003:**
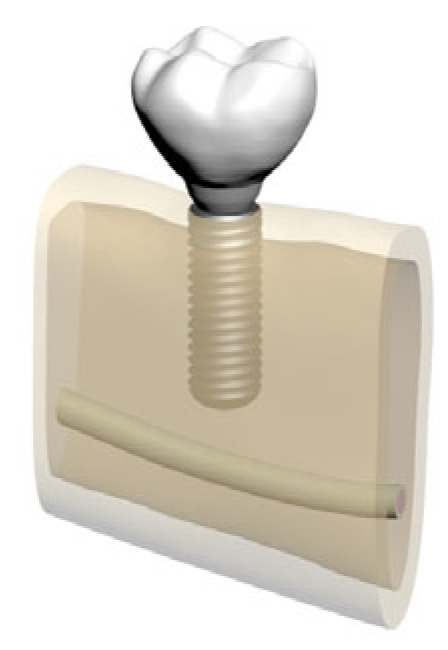
Model 02: Normal mandible (D2), 10 mm bone width, 12 mm implant.

**Figure 4 bioengineering-12-00888-f004:**
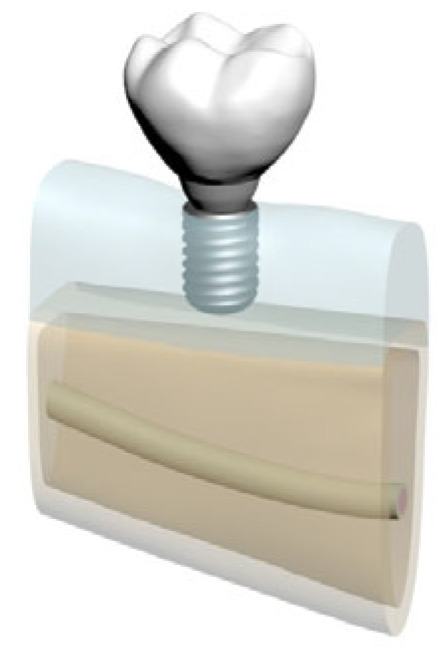
Model 03: Grafted mandible (D3), 8 mm bone width, 6 mm implant.

**Figure 5 bioengineering-12-00888-f005:**
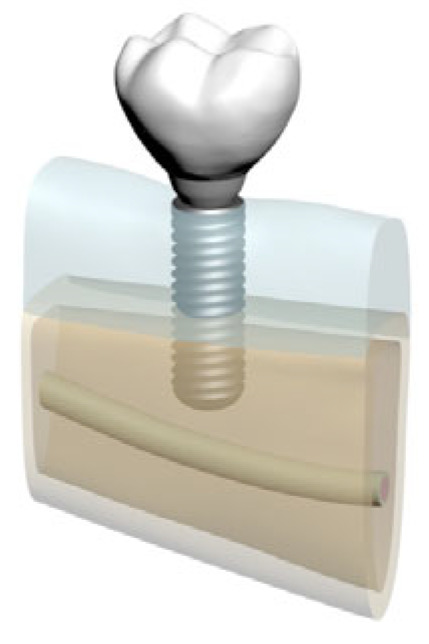
Model 04: Grafted mandible (D3), 8 mm bone width, 12 mm implant.

**Figure 6 bioengineering-12-00888-f006:**
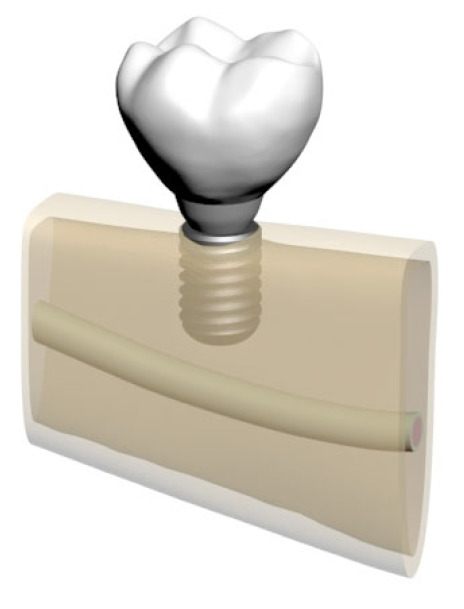
Model 05: Atrophic mandible (D4), 6 mm bone width, 6 mm implant.

**Figure 7 bioengineering-12-00888-f007:**
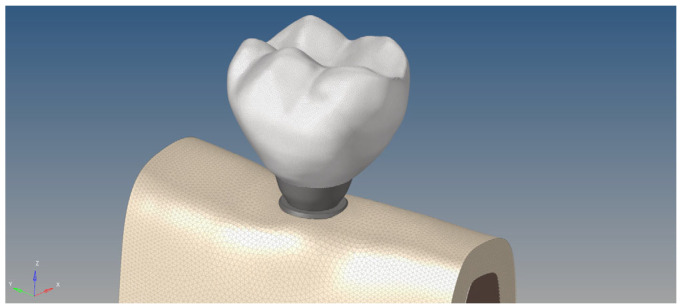
Tetrahedral mesh representation of anatomical models.

**Figure 8 bioengineering-12-00888-f008:**
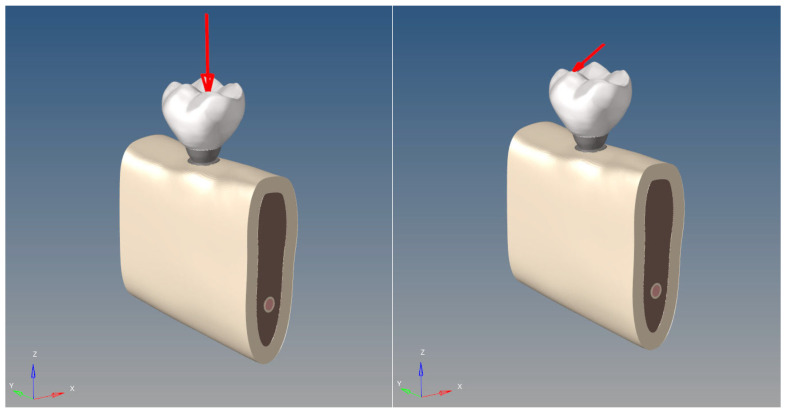
Loading scenarios. The left image shows a vertical load of 200 N, and the right image shows a 30° oblique load of 100 N (arrows indicate the direction of the applied load).

**Figure 9 bioengineering-12-00888-f009:**
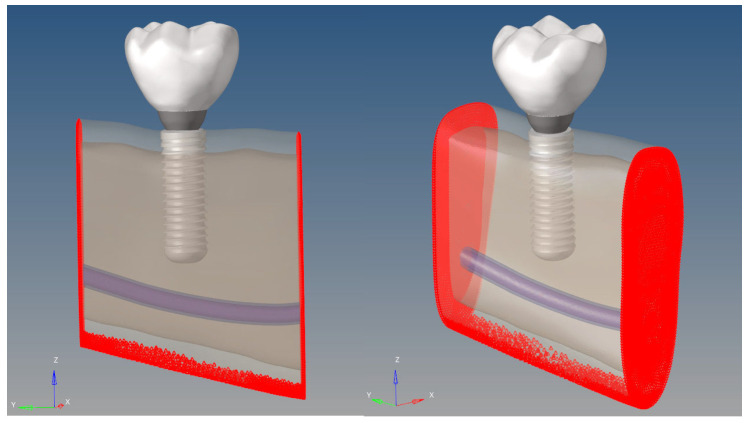
Boundary conditions. (The red areas indicate the fixed boundary conditions and the transparent volume shows the bone domain.).

**Figure 10 bioengineering-12-00888-f010:**
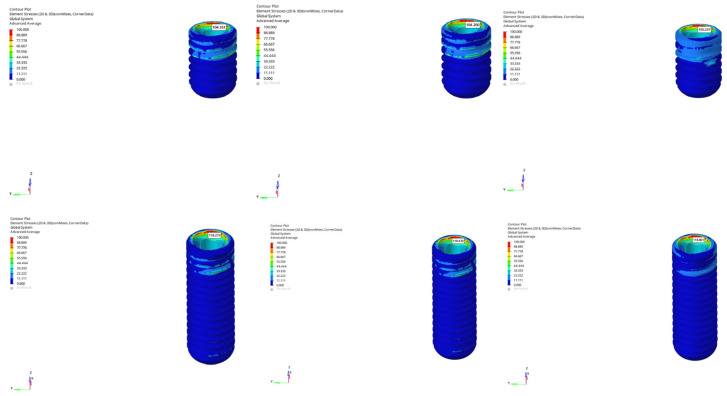
von Mises stress distributions in the implant body for 6 mm and 12 mm implants in normal mandible under immediate, early, and delayed loading (from **left** to **right**).

**Figure 11 bioengineering-12-00888-f011:**
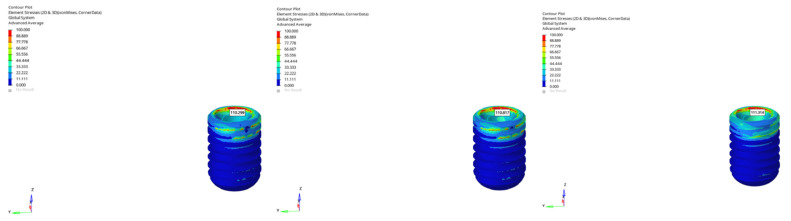
von Mises stress distributions in the implant body for 6 mm implant in atrophic mandible under immediate, early, and delayed vertical loading (from **left** to **right**).

**Figure 12 bioengineering-12-00888-f012:**
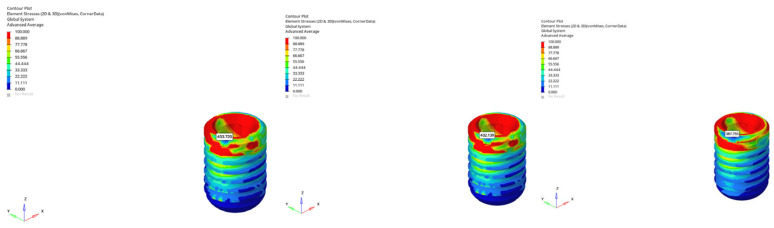
von Mises stress distributions in the implant body for 6 mm implant in atrophic mandible under immediate, early, and delayed oblique loading (from **left** to **right**).

**Figure 13 bioengineering-12-00888-f013:**
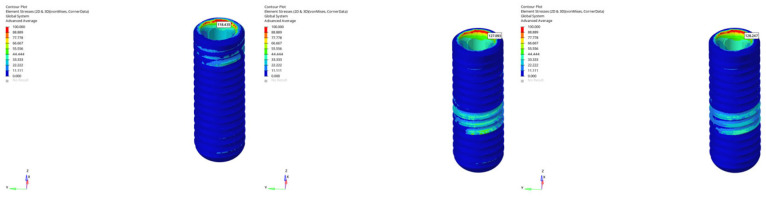
von Mises stress distributions in the implant body for 12 mm implant in grafted atrophic mandible under immediate, early, and delayed vertical loading (from **left** to **right**).

**Figure 14 bioengineering-12-00888-f014:**
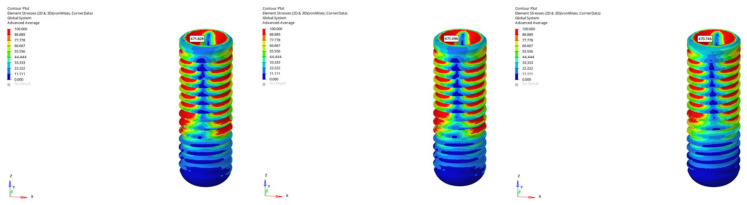
von Mises stress distributions in the implant body for 12 mm implant in grafted atrophic mandible under immediate, early, and delayed oblique loading (from **left** to **right**).

**Figure 15 bioengineering-12-00888-f015:**
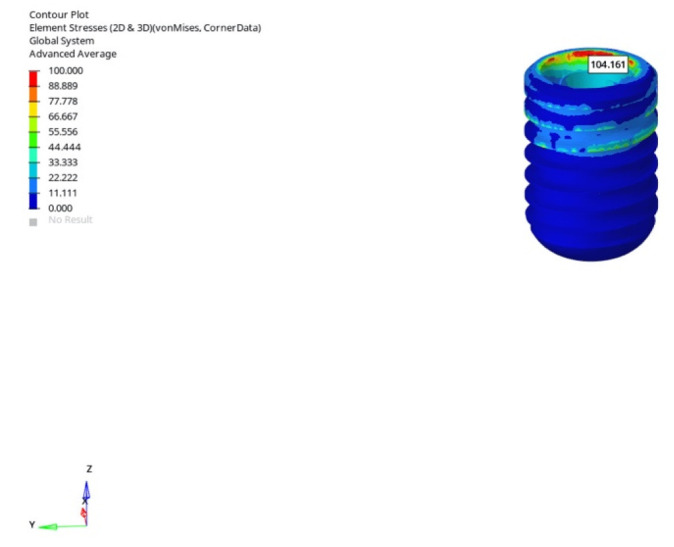
The lowest von Mises stress in the implant body (104.161 MPa) was recorded under vertical loading (0°) with a 6 mm implant and the immediate loading condition in the normal mandible model. The vertical force was applied to the central fossa of the crown, and boundary constraints were defined at the superior and inferior cortical bone regions to simulate fixation.

**Figure 16 bioengineering-12-00888-f016:**
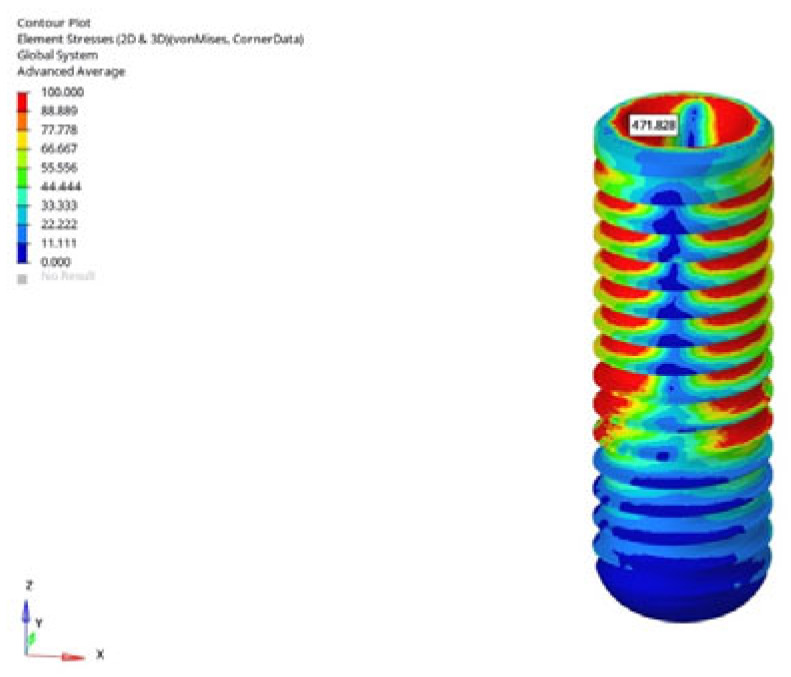
The highest von Mises stress in the implant body (471.828 MPa) was obtained under oblique loading (30°) with a 12 mm implant and the immediate loading condition in the grafted atrophic mandible model. The oblique load was applied to the lingual slope of the buccal cusp, and boundary constraints were applied at the inferior and superior cortical regions of the mandible.

**Figure 17 bioengineering-12-00888-f017:**
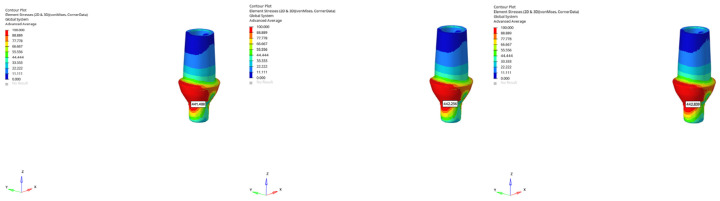
von Mises stress distribution at the abutment level for the 6 mm implant in normal mandible under oblique loading (from **left** to **right**: immediate, early, and delayed loading).

**Figure 18 bioengineering-12-00888-f018:**
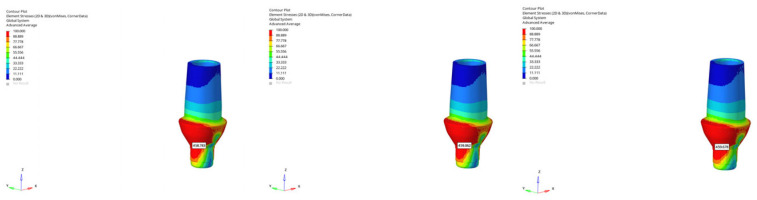
von Mises stress distribution at the abutment level for the 12 mm implant in normal mandible under oblique loading (from **left** to **right**: immediate, early, and delayed loading).

**Figure 19 bioengineering-12-00888-f019:**
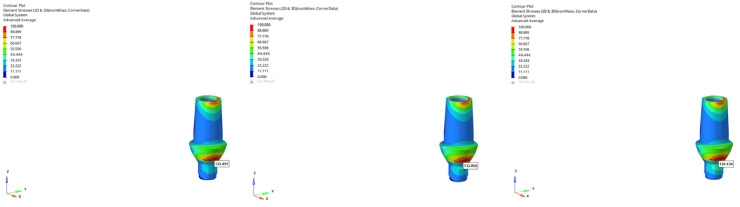
von Mises stress distribution for the 6 mm implant in atrophic mandible under vertical loading (from **left** to **right**: immediate, early, and delayed loading).

**Figure 20 bioengineering-12-00888-f020:**
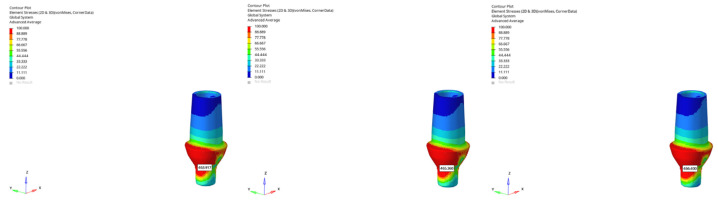
von Mises stress distribution for the 6 mm implant in atrophic mandible under oblique loading (from **left** to **right**: immediate, early, and delayed loading).

**Figure 21 bioengineering-12-00888-f021:**
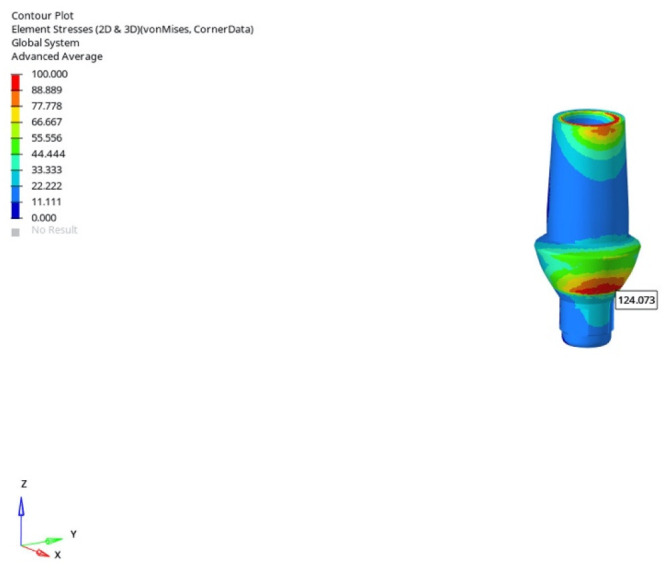
The lowest abutment von Mises stress (124.073 MPa) obtained in the normal mandible model under vertical loading (0°) using a 6 mm implant and the immediate loading condition.

**Figure 22 bioengineering-12-00888-f022:**
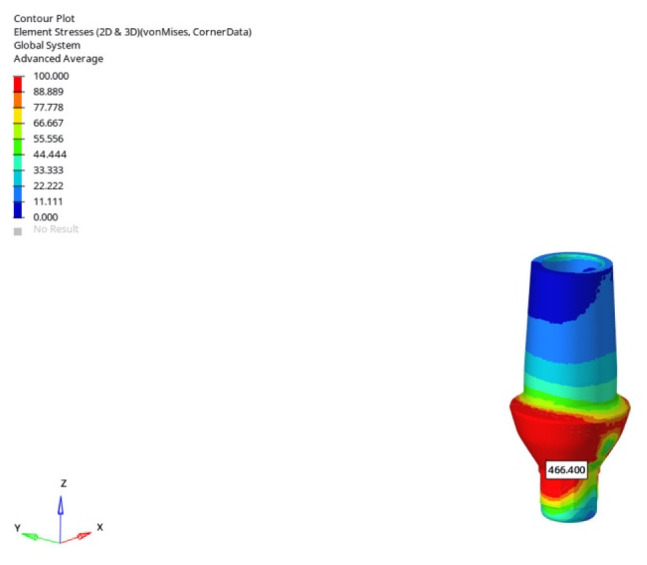
The highest abutment von Mises stress (466.400 MPa) obtained under oblique loading (30°) with a 6 mm implant and the delayed loading condition in the atrophic mandible model.

**Figure 23 bioengineering-12-00888-f023:**
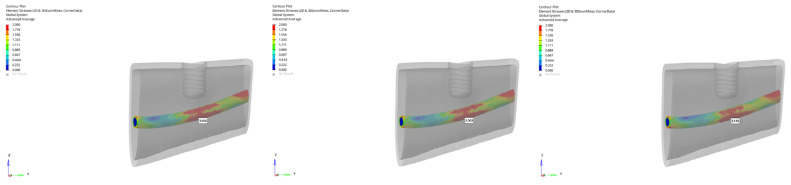
von Mises stress distribution around the mandibular canal for the 6 mm implant under vertical loading (from **left** to **right**: immediate, early, and delayed loading).

**Figure 24 bioengineering-12-00888-f024:**
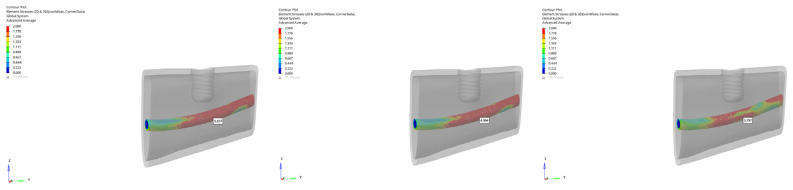
von Mises stress distribution around the mandibular canal for the 6 mm implant under oblique loading (from **left** to **right**: immediate, early, and delayed loading).

**Figure 25 bioengineering-12-00888-f025:**
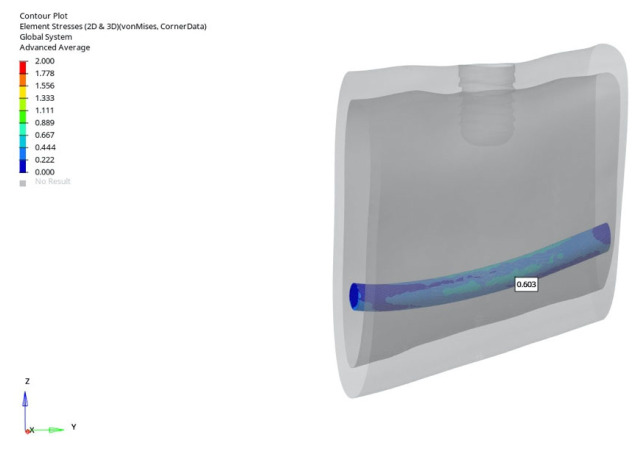
The lowest von Mises stress around the mandibular canal (0.603 MPa) was recorded in the normal mandible model under oblique force (30°) using a 6 mm implant with late loading conditions.

**Figure 26 bioengineering-12-00888-f026:**
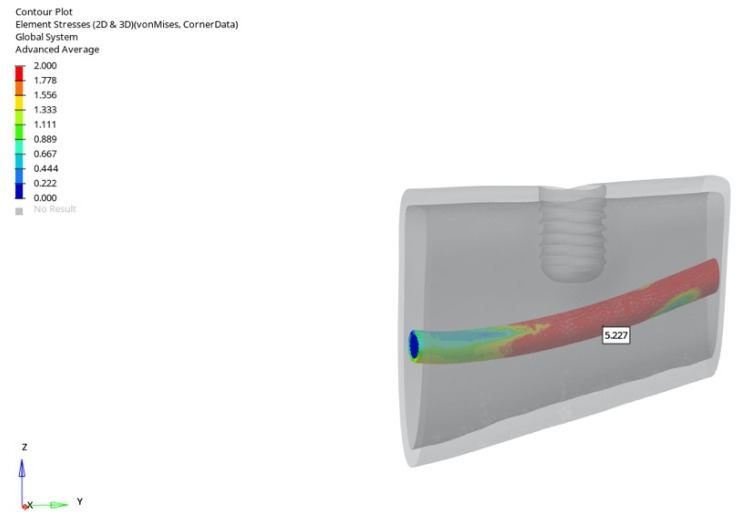
The highest von Mises stress around the mandibular canal (5.227 MPa) was recorded in the atrophic mandible model under oblique force (30°) using a 6 mm implant with immediate loading conditions.

**Figure 27 bioengineering-12-00888-f027:**
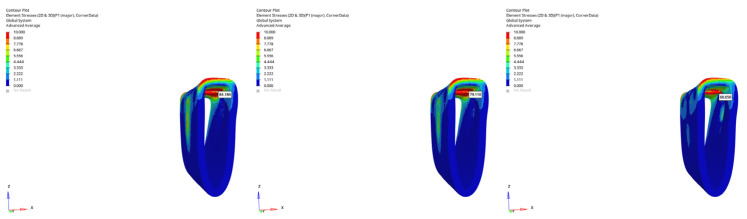
Maximum (tensile) principal stresses at cortical bone point P1 under vertical loading for the 6 mm implant in the atrophic mandible (**left** to **right**: immediate, early, delayed loading).

**Figure 28 bioengineering-12-00888-f028:**
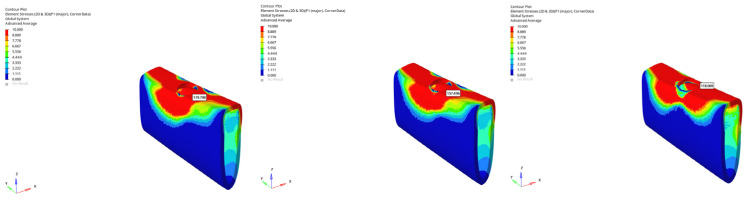
Maximum (tensile) principal stresses at cortical bone point P1 under oblique loading for the 6 mm implant in the atrophic mandible (**left** to **right**: immediate, early, delayed loading).

**Figure 29 bioengineering-12-00888-f029:**
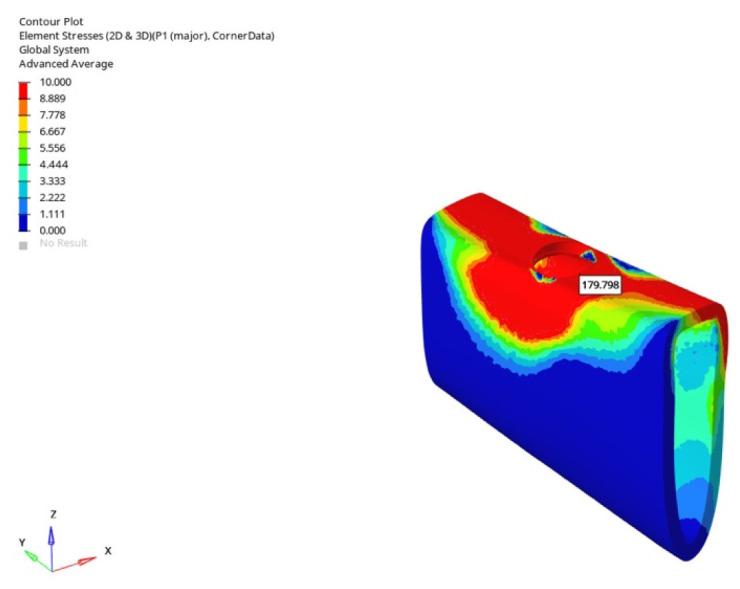
Maximum tensile stress at the cortical P1 point (179.798 MPa) obtained under oblique force (30°) with a 6 mm implant and the immediate loading condition in the atrophic mandible model.

**Figure 30 bioengineering-12-00888-f030:**
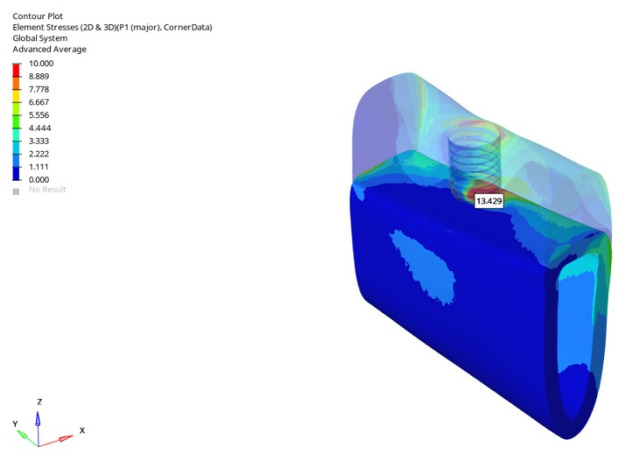
In the grafted atrophic mandible model, under oblique force (30°), the lowest cortical P1 maximum tensile stress (13.429 MPa) was obtained with a 6 mm implant and the delayed loading condition.

**Figure 31 bioengineering-12-00888-f031:**
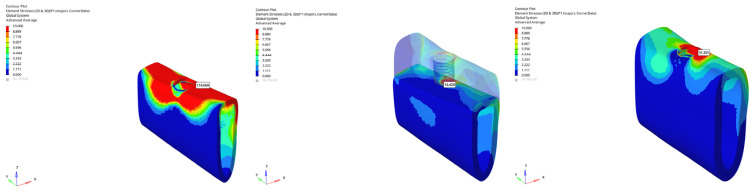
Maximum (tensile) principal stresses at cortical bone point P1 under oblique loading in the delayed loading condition for the 6 mm implant (**left** to **right**: atrophic, grafted atrophic, and normal mandible).

**Figure 32 bioengineering-12-00888-f032:**
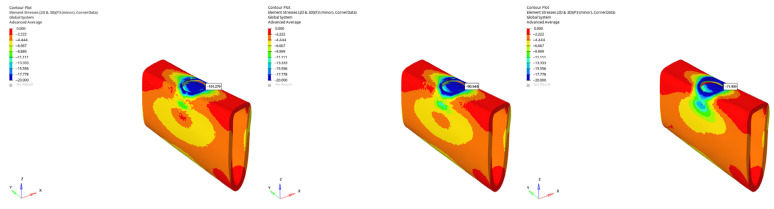
Maximum (compressive) principal stresses at point P3 in cortical bone under vertical loading in atrophic mandible for the 6 mm implant (**left** to **right**: immediate, early, and delayed loading conditions).

**Figure 33 bioengineering-12-00888-f033:**
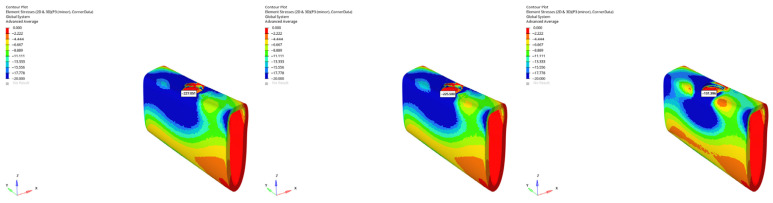
Maximum (compressive) principal stresses at point P3 in cortical bone under oblique loading in atrophic mandible for the 6 mm implant (**left** to **right**: immediate, early, and delayed loading conditions).

**Figure 34 bioengineering-12-00888-f034:**
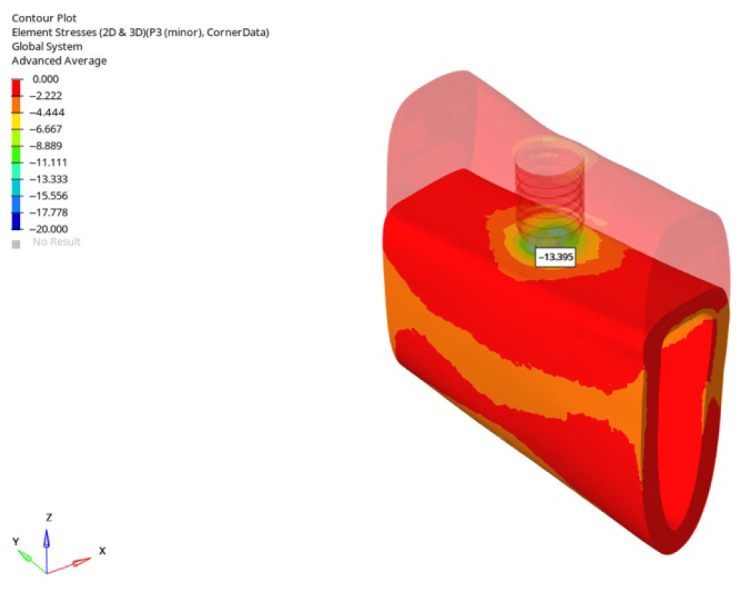
In the grafted atrophic mandible model, under vertical force (0°), with a 6 mm implant and the delayed loading condition, the lowest compressive stress (−13.395 MPa) was recorded in the cortical P3 region.

**Figure 35 bioengineering-12-00888-f035:**
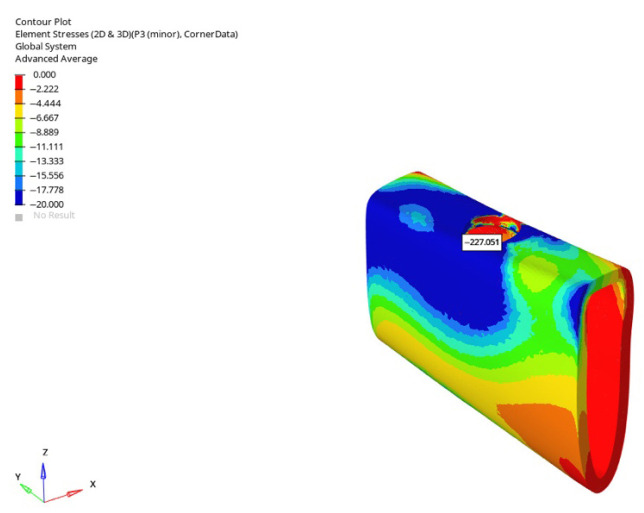
In the atrophic mandible model, under oblique force (30°), with a 6 mm implant and the immediate loading condition, the highest compressive stress (−227.051 MPa) occurred in the cortical P3 region.

**Figure 36 bioengineering-12-00888-f036:**
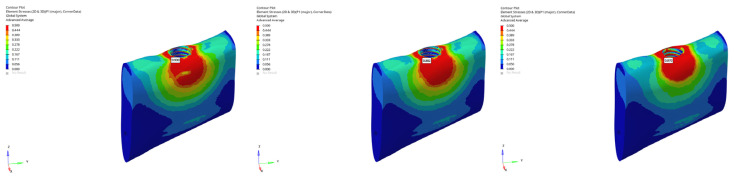
Maximum (tensile) principal stresses at the P1 point in trabecular bone under vertical loading in normal mandible for the 6 mm implant (**left** to **right**: immediate, early, and delayed loading conditions).

**Figure 37 bioengineering-12-00888-f037:**
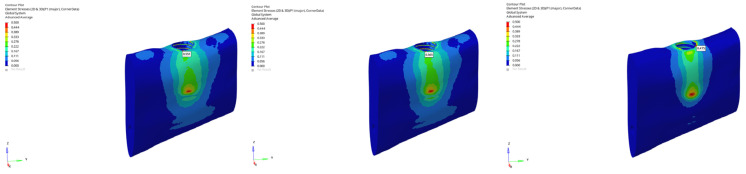
Maximum (tensile) principal stresses at the P1 point in trabecular bone under vertical loading in normal mandible for the 12 mm implant (**left** to **right**: immediate, early, and delayed loading conditions).

**Figure 38 bioengineering-12-00888-f038:**
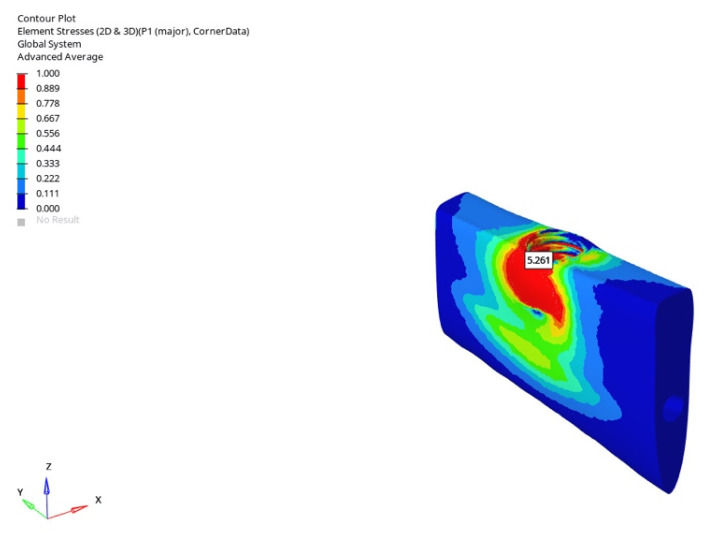
The highest trabecular P1 maximum tensile stress (5.261 MPa) was obtained in the atrophic mandible model under oblique force (30°) with a 6 mm implant and immediate loading conditions.

**Figure 39 bioengineering-12-00888-f039:**
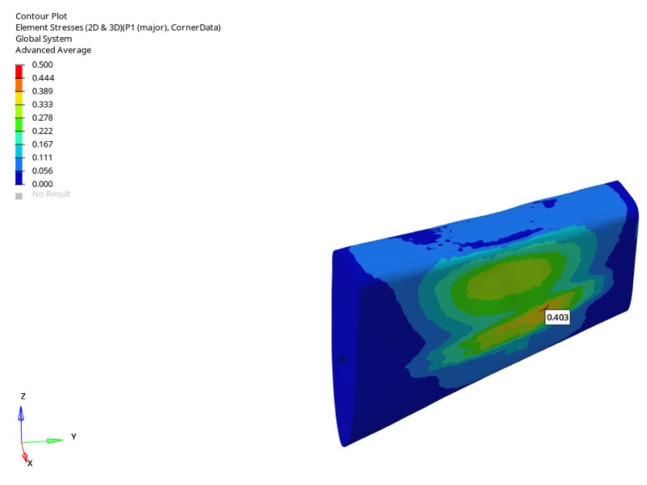
In the grafted atrophic mandible model, under vertical force (0°), the lowest trabecular P1 maximum tensile stress (0.403 MPa) was obtained with a 6 mm implant and delayed loading conditions.

**Figure 40 bioengineering-12-00888-f040:**
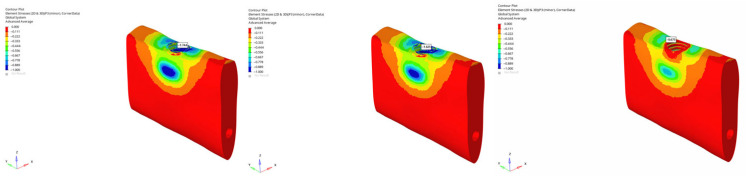
Minimum (compressive) principal stresses at the P3 point in trabecular bone under vertical loading in normal mandible for the 6 mm implant (**left** to **right**: immediate, early, and delayed loading conditions).

**Figure 41 bioengineering-12-00888-f041:**
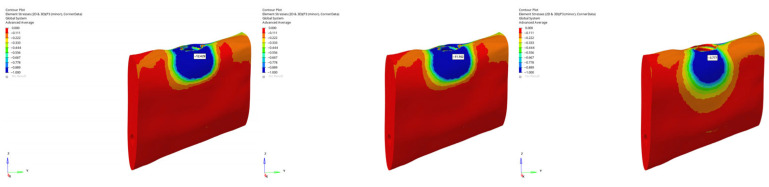
Minimum (compressive) principal stresses at the P3 point in trabecular bone under oblique loading in normal mandible for the 6 mm implant (**left** to **right**: immediate, early, and delayed loading conditions).

**Figure 42 bioengineering-12-00888-f042:**
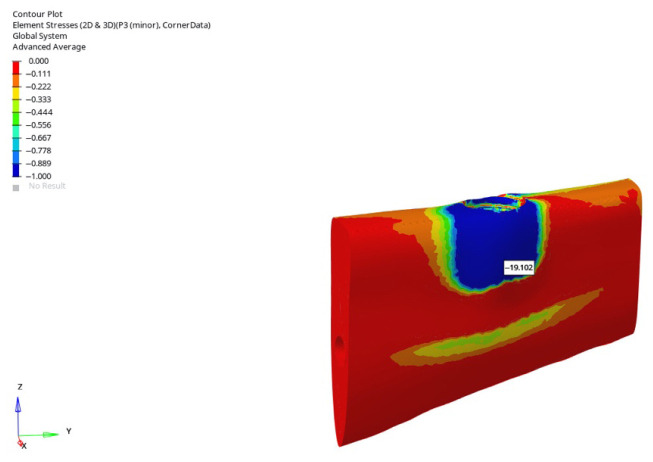
In the atrophic mandible model, under oblique force (30°), the highest trabecular P3 compressive stress (−19.102 MPa) was obtained with a 6 mm implant and the immediate loading condition.

**Figure 43 bioengineering-12-00888-f043:**
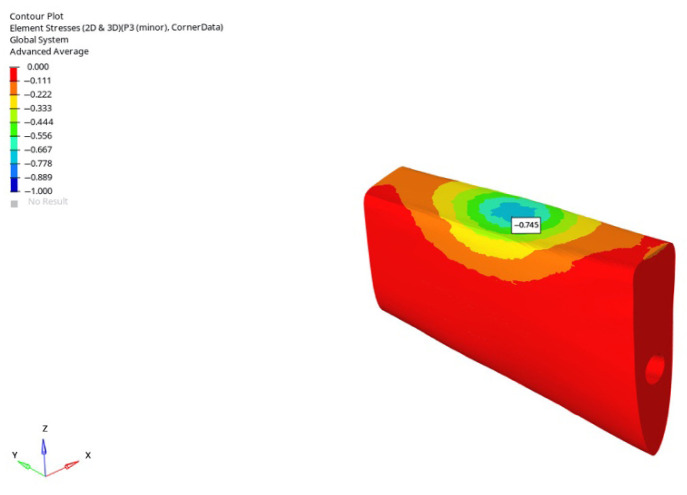
In the grafted atrophic mandible model, under vertical force (0°), with a 6 mm implant and the delayed loading condition, the lowest trabecular P3 compressive stress (−0.745 MPa) was recorded.

**Table 1 bioengineering-12-00888-t001:** Morphometric measurements of the mandible.

Structural Feature	Normal (D2)	Grafted (D3)	Atrophic (D4)
Superior cortical bone (mm)	2.0	1.5	1.0
Superior trabecular bone (mm)	13.0	12.5	7.0
Inferior cortical bone (mm)	2.0	1.5	1.0
Inferior trabecular bone (mm)	3.0	3.5	4.0
Crest–canal top distance (mm)	15.0	14.0	8.0
Canal bottom–base distance (mm)	5.0	5.0	5.0
Total bone width (mm)	10.0	8.0	6.0

**Table 2 bioengineering-12-00888-t002:** Node and element counts of analysis models.

Model	Node Count	Element Count
Model 01	227,757	922,400
Model 02	266,980	1,077,669
Model 03	215,050	866,050
Model 04	259,782	1,039,637
Model 05	191,848	762,723

**Table 3 bioengineering-12-00888-t003:** von Mises stresses and peri-implant bone stresses under vertical loading (0°) in the normal mandible.

Loading Scenario	Implant Length	VM Implant (MPa)	VM Abutment (MPa)	VM Mandibular Canal (MPa)	Cortical P1 (MPa)	Cortical P3 (MPa)	Trabecular P1 (MPa)	Trabecular P3 (MPa)
Immediate	6 mm	104.161	124.073	0.937	28.335	−28.180	0.930	−1.744
	12 mm	118.273	127.291	1.827	17.949	−21.025	0.551	−1.254
Early	6 mm	104.260	124.096	0.933	28.094	−27.206	0.882	−1.625
	12 mm	118.435	127.544	1.800	16.581	−20.645	0.565	−1.251
Delayed	6 mm	105.237	124.594	0.865	22.956	−25.275	0.870	−0.675
	12 mm	118.801	128.319	1.765	15.519	−20.550	0.475	−0.525

**Table 4 bioengineering-12-00888-t004:** von Mises stresses and peri-implant bone stresses under oblique loading in the normal mandible.

Loading Scenario	Implant Length	VM Implant (MPa)	VM Abutment (MPa)	VM Mandibular Canal (MPa)	Cortical P1 (MPa)	Cortical P3 (MPa)	Trabecular P1 (MPa)	Trabecular P3 (MPa)
Immediate	6 mm	365.352	441.488	0.768	72.730	−87.935	4.792	−12.429
	12 mm	465.698	458.783	1.133	44.052	−52.519	1.852	−10.430
Early	6 mm	364.862	442.256	0.732	70.227	−85.808	4.135	−11.162
	12 mm	465.541	459.062	1.129	44.039	−52.447	1.653	−9.747
Delayed	6 mm	346.366	442.839	0.603	51.201	−56.861	4.123	−3.777
	12 mm	452.020	459.678	1.065	43.731	−44.907	1.549	−2.705

**Table 5 bioengineering-12-00888-t005:** von Mises stresses and peri-implant bone stresses under vertical loading in the atrophic mandible.

Loading Scenario	Implant Length	VM Implant (MPa)	VM Abutment (MPa)	VM Mandibular Canal (MPa)	Cortical P1 (MPa)	Cortical P3 (MPa)	Trabecular P1 (MPa)	Trabecular P3 (MPa)
Immediate	6 mm	110.299	132.491	3.656	84.186	−101.279	1.581	−2.328
Early	6 mm	110.817	132.856	3.503	79.110	−90.546	1.467	−1.824
Delayed	6 mm	111.314	134.434	3.146	60.650	−71.909	1.411	−1.635

**Table 6 bioengineering-12-00888-t006:** von Mises stresses and peri-implant bone stresses under oblique loading in the atrophic mandible.

Loading Scenario	Implant Length	VM Implant (MPa)	VM Abutment (MPa)	VM Mandibular Canal (MPa)	Cortical P1 (MPa)	Cortical P3 (MPa)	Trabecular P1 (MPa)	Trabecular P3 (MPa)
Immediate	6 mm	403.720	463.917	5.227	179.798	−227.051	5.261	−19.102
Early	6 mm	402.139	465.360	4.994	157.496	−225.500	5.257	−17.006
Delayed	6 mm	387.755	466.400	3.797	118.069	−131.396	5.069	−6.797

**Table 7 bioengineering-12-00888-t007:** von Mises stresses and peri-implant bone stresses under vertical loading in the grafted atrophic mandible.

Loading Scenario	Implant Length	VM Implant (MPa)	VM Abutment (MPa)	VM Mandibular Canal (MPa)	Cortical P1 (MPa)	Cortical P3 (MPa)	Trabecular P1 (MPa)	Trabecular P3 (MPa)
Immediate	6 mm	104.278	127.166	1.891	16.391	−14.471	0.409	−0.786
	12 mm	118.435	127.359	2.480	16.828	−25.131	0.634	−1.508
Early	6 mm	104.656	127.168	1.891	16.174	−14.043	0.405	−0.780
	12 mm	127.093	127.629	2.426	16.675	−24.323	0.567	−1.500
Delayed	6 mm	105.625	127.189	1.888	15.221	−13.395	0.403	−0.745
	12 mm	128.247	128.640	2.287	15.485	−23.007	0.514	−1.484

**Table 8 bioengineering-12-00888-t008:** von Mises stresses and peri-implant bone stresses under oblique loading in the grafted atrophic mandible.

Loading Scenario	Implant Length	VM Implant (MPa)	VM Abutment (MPa)	VM Mandibular Canal (MPa)	Cortical P1 (MPa)	Cortical P3 (MPa)	Trabecular P1 (MPa)	Trabecular P3 (MPa)
Immediate	6 mm	392.595	447.861	2.171	15.832	−38.615	0.591	−0.869
	12 mm	471.828	458.023	2.971	32.591	−17.908	1.115	−6.715
Early	6 mm	391.753	448.331	2.140	14.728	−36.749	0.572	−0.851
	12 mm	471.496	458.262	2.934	25.856	−17.629	1.081	−5.548
Delayed	6 mm	389.284	449.454	2.051	13.429	−21.010	0.539	−0.810
	12 mm	470.746	459.382	2.263	21.432	−15.442	0.977	−5.138

## Data Availability

The data supporting the findings of this study are available from the corresponding author upon reasonable request.

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
