# Peer review of "Effect of Bone Quality, Implant Length, and Loading Timing on Stress Transmission in the Posterior Mandible: A Finite Element Analysis"

_bioengineering, 2025, doi:10.3390/bioengineering12080888_

Round 1
Reviewer 1 Report
Comments and Suggestions for Authors
COMMENTS TO THE AUTHORS – REQUIRED REVISIONS
ABSTRACT
Please specify the numerical values of both axial and oblique loading applied in the simulations.
Clearly state the criteria used to analyze the results (e.g., maximum principal stresses, von Mises stresses, stress concentrations around the implant).
Avoid using only percentages in the results. For instance, instead of stating: “maximum stress levels in trabecular bone decreased by approximately 40–50%,” include the actual stress range.
Revise the abstract to include the key stress ranges observed in different models and scenarios.
In the sentence “These findings indicate that oblique loading and reduced bone volume may compromise implant success”:
Replace “implant success” with “implant survival”, since the former refers to a clinical concept not fully addressed here.
Emphasize the relevance of graft application in more detail, include donor regions, what was simulated, and its biomechanical impact. Highlight the novelty of this aspect in comparison to the already well-established literature on implant length.
INTRODUCTION
Include references in the first paragraph to support the general context and motivation of the study.
Formulate a clear hypothesis at the end of the Introduction to guide the study rationale and allow its discussion later.
METHODS
Since tomographic data from the Visible Human Project were used, please clarify whether ethical approval was required or exempted due to the use of public anatomical datasets.
In Table 1, cite the source or justification for the bone quality classification system (D2, D3, D4).
In Section 2.3 (Material properties), provide references for each elastic modulus and Poisson’s ratio adopted.
The sentence “The forces were distributed evenly across multiple nodes to prevent stress singularities…” needs more detail. How many nodes were involved? Include an illustrative figure showing the loading region and mesh.
The methodology includes ten linear static and twenty nonlinear static analyses. This is an important innovation and should be emphasized:
Justify the rationale behind running so many simulations.
Most FEA studies in the field report only one or two simulations; discuss how your approach improves accuracy or robustness.
RESULTS
Correct the terminology: write “von Mises” (not “Von Mises”).
In Table 3, explain how the stress values were derived:
Are they averages?
Over what region or volume around the implant were these values measured?
The statement “The lowest compressive stress was –13.395 MPa [...] the highest was –227.051 MPa” should be further discussed:
Do these values exceed the bone’s strength thresholds?
What are the implications for clinical relevance?
Avoid showing only one representative image per group:
Present panels of all simulated conditions (e.g., implant length, loading type, bone condition).
Add standardized illustrations of the models, showing meshing, loading conditions, and boundary constraints for each scenario.
DISCUSSION
Reiterate the study hypothesis (to be stated in the Introduction) and discuss whether it was supported or rejected.
Include a limitations subsection, addressing:
Simplifications in geometry or material properties,
Lack of dynamic loading,
Limitations of linear vs. nonlinear modeling, etc.
The conclusions section should be succinct and itemized. The last paragraph currently in the conclusions would fit better in the Discussion.
Author Response
Please see the attachment, thank you..

Reviewer 2 Report
Comments and Suggestions for Authors
This study presents a well-structured finite element analysis (FEA) evaluating the effects of implant length, bone quality, grafting, and loading timing on peri-implant stress distribution in the posterior mandible. The model design, material properties, and loading protocols are generally appropriate. However, several issues require clarification and correction before publication. These points are focused on methodological limitations, inconsistencies in interpretation, and missing descriptions of modeling assumptions.
Although the study emphasizes that longer implants reduce stress in surrounding bone, Table 3 and Table 4 show that Von Mises stress in the implant body is consistently higher in 12 mm implants than in 6 mm implants. This contradicts the generalized conclusion that longer implants are universally biomechanically favorable. The authors should clarify that while stress in bone may be reduced, stress within the implant itself increases with length—potentially increasing mechanical failure risks.
In Figures 3 and 4 and Tables 4, 6, and 8, abutment stress reaches as high as 466 MPa under oblique loading. These values are significantly elevated and may exceed the mechanical fatigue thresholds for implant-abutment connections. However, the discussion section offers little to no mention of potential complications such as screw loosening or abutment fracture. This omission should be addressed, particularly when highlighting the implications of implant length.
All materials were modeled as isotropic, linear-elastic, and homogeneous. While this is a common FEA simplification, it is especially problematic for Bio-Oss graft material, which is known to be porous and anisotropic. Similarly, bone exhibits anisotropic mechanical behavior, especially between cortical and trabecular regions. A brief discussion of this limitation should be included in the manuscript.
Only one oblique loading scenario at 30° was modeled. This may not fully represent the range of occlusal vectors present in posterior mandibular function. The limitation of applying only a single angled force should be acknowledged.
The authors modeled different osseointegration stages using friction coefficients of 0.3 (immediate), 0.5 (early), and “FREEZE” (delayed). However, no rationale or literature reference is provided for these values. As these directly impact stress distribution, the basis for these assumptions should be explained.
The manuscript states that all degrees of freedom were constrained at selected nodes to prevent displacement. This may overconstrain the model, particularly since real mandibular bone exhibits elastic deformation. A brief explanation of how this simplification might affect stress results should be included.
The vertical and oblique forces are described as being “distributed across nodes,” but no information is given on the number of nodes or surface area involved. For reproducibility and clarity, this should be more explicitly defined.
The manuscript does not report any convergence analysis, mesh sensitivity test, or validation through experimental data or previous benchmarks. While FEA is a theoretical method, the validity of simulation results depends on confirming that they are mesh-independent and reflective of real mechanical behavior. This absence should be mentioned as a limitation.
Author Response
Please see the attachment,thank you..
